# "I think they are infected because of their ignorance and lack of responsibility": A mixed-methods study on HIV-related stigma in the healthcare system in Kazakhstan

Balnur Iskakova[1]*, Elizabeth J. King[2], Recai Murat Yucel[3], Jack DeHovitz[4], Zhamilya Nugmanova[1]

**1** Department of Epidemiology, School of Public Health, Kazakh National Medical University named after S.D, Asfendiyarov, Almaty, Kazakhstan, **2** Department of Health Behavior and Health Equity, School of Public Health, University of Michigan, Ann Arbor, Michigan, United States of America, **3** Department of Epidemiology and Biostatistics, Temple University College of Public Health, Philadelphia, Pennsylvania, United States of America, **4** SUNY Downstate Health Sciences University, Brooklyn, New York, United States of America

* balnurkskak@gmail.com

## Abstract

### Background

HIV-related stigma among healthcare providers remains a significant barrier to effective HIV care and treatment. This study aimed to assess HIV-related stigma and associated factors among healthcare providers in primary healthcare clinics (PHCs) in Almaty, Kazakhstan.

### Methods

A mixed-methods design was employed, involving quantitative surveys and qualitative in-depth interviews. Eight PHCs were randomly selected, and 448 healthcare providers participated in the survey. The cross-sectional surveys took place at the PHCs between May 2, 2019, and July 2, 2019. The sample size was predetermined statistically with a 2.5% precision using a 95% confidence level. For the qualitative component, 10 participants were chosen for in-depth interviews. Descriptive statistics, bivariate analysis, and multivariable logistic regression models were performed for quantitative data analysis. Qualitative data were analyzed through manual thematic analysis.

### Results

The study revealed high levels of HIV-related stigma, with 87% of respondents agreeing with at least one stigmatizing statement about people living with HIV (PLHIV). Fear of HIV infection was also prevalent, with 85% of healthcare providers expressing some

**Data availability statement:** All relevant data are within the manuscript and its Supporting Information files.

**Funding:** The author(s) received no specific funding for this work.

**Competing interests:** The authors have declared that no competing interests exist.

level of concern about contracting HIV during medical procedures. Logistic regression analysis indicated that longer years of work in healthcare were protective against stigmatizing opinions (Adjusted Odds Ratio (AOR)=0.25; 95% Confidence Interval (95%CI)=0.09,0.67; p=0.006), while not having seen a patient living with HIV in the last 12 months was associated with higher stigma (AOR=3.31; 95%CI=1.73, 6.31; p<0.001). Qualitative interviews corroborated these findings and highlighted differential attitudes towards PLHIV based on modes of transmission, with particularly negative views towards sex workers and individuals with non-traditional sexual orientations.

## Conclusions

The study demonstrates significant HIV-related stigma among healthcare providers in Almaty PHCs, influenced by a lack of exposure to PLHIV and specific socio-demographic factors. These findings underscore the need for targeted interventions to reduce stigma and enhance HIV care in Kazakhstan.

## Introduction

### HIV in Kazakhstan

Despite advances made in the clinical management of HIV, 24% of people living with HIV (PLHIV) worldwide do not have access to antiretroviral therapy (ART), many receive the initial treatment at late stages, and around 680,000 PLHIV die from AIDS-related illnesses each year [1,2]. The HIV epidemic in Kazakhstan, a country in the Eastern European and Central Asian (EECA) region, continues to increase, in contrast to other regions. There were 40,000 PLHIV in Kazakhstan by the latest estimates in 2023, and the highest prevalence of the infection is documented in several regions, including Pavlodar, Karaganda, Almaty, North Kazakhstan, Kostanay, and East Kazakhstan [3,4]. Although Kazakhstan has made progress, it has not yet achieved the 90-90-90 targets [5]. Following the Joint United Nations Program on ending AIDS, among PLHIV in Kazakhstan, 82% are aware of their status, 68% of those diagnosed are on ART, and 78% of those receiving ART have achieved viral suppression [5]. Thus, for the country to reach the more ambitious 95-95-95 goals by 2025, substantial efforts are required, particularly in addressing stigma and discrimination, improving access to HIV testing and treatment, and strengthening the healthcare system [6].

The HIV epidemic in Kazakhstan is concentrated predominantly among key populations and continues to rise [6,7]. Notably, HIV incidence in Kazakhstan surged by 88% from 2010 to 2021, making it the seventh highest increase globally, while the number of people living with HIV more than doubled [7]. The prevalence rates among key affected populations, such as people who inject drugs (PWID), men who have sex with men (MSM), prisoners, and sex workers (SWs), were 7.60%, 8.80%, 4.20%, and 1.50%, respectively, in 2023 [8]. Recruiting such groups into HIV treatment can be challenging, due in part to the amount of stigma and discrimination they experience in their daily lives.

## Understanding the concept of stigma

HIV is one of the world's highly stigmatized conditions. The term 'stigma' is associated with "exclusion", "rejection", or "blame" due to holding a particular condition, which is a subject for social judgment [9]. HIV-related stigma is described by Goffman as "a process of devaluation of people either living with, or associated with, HIV and AIDS" [9]. Stigma associated with HIV/AIDS can also be internalized by PLHIV, meaning that the general negative reactions to HIV from others may result in "internal stigma" among PLHIV [10]. This type of stigma is differentiated from "enacted stigma," which is the experience of unfair treatment by others. Internal stigma may lead to low self-esteem, self-imposed withdrawal, and even suicidal feelings, while enacted stigma may exacerbate such feelings, resulting in an unwillingness to engage with others [10–12]. Fear of disclosure and shame caused by HIV-related stigma have been shown to keep PLHIV from getting the care needed in time [11].

## HIV-related stigma in healthcare

HIV-related stigma develops when one starts making assumptions about a person's HIV status based on personal characteristics. For example, healthcare providers may assume that only certain kinds of people are at risk or that some people are less worthy of care than others due to their lifestyles. Studies on HIV-related stigma conducted in China and Vietnam suggest that healthcare staff may adopt specific attitudes and behaviors to align with their peers and gain social acceptance [13,14]. Furthermore, there is evidence that stigmatizing interactions are not even recognized by healthcare workers as stigmatizing [15]. Stigmatizing behaviors can be unintentionally conducted and perceived negatively. Although some healthcare providers in resource-limited settings may prefer to visibly mark the files of PLHIV for perceived safety reasons, this practice can lead to breaches of patient confidentiality and reinforce stigma [15].

PLHIV in Kazakhstan receive HIV care at the Centers for AIDS Prevention and Control (AIDS Centers). The main AIDS Centers are located in the largest cities of the country, such as Almaty, Astana, and Shymkent. The focus of such centers is to provide diagnostic measures and treatment of HIV, including prevention activities and epidemiological monitoring [16]. HIV care is free of charge to Kazakhstani citizens, and starting from May 2017, all confirmed cases of HIV can get ART regardless of one's CD4 count [16]. These centers are separate from primary healthcare settings (PHCs) both by the type of care they provide and the infectious disease focus of the healthcare staff.

Good access to care in primary and secondary care settings and communication between these services itself is crucial in managing HIV. Patients may feel more comfortable and develop trusting relationships in supportive healthcare environments [17]. Earlier studies conducted on HIV-related stigma in healthcare settings in Kazakhstan were restricted to small sample size studies and/or focused on patients' perspectives only on addressing this issue [6,18,19]. This exploratory study aimed to assess HIV-related stigma in Kazakhstani primary healthcare settings and the factors leading to stigmatization of PLHIV using a mixed-methods approach.

## Materials and methods

### Study setting

This study was conducted in the city of Almaty, which has the highest prevalence of HIV after the Eastern Kazakhstan region, Karaganda, and Pavlodar. There are 65 PHCs in Almaty, located evenly among 8 districts (Alatau, Almaly, Auzeov, Bostandyk, Zhetysu, Medeu, Nauryzbai, and Turksib). The general public, including PLHIV, can access necessary medical care in PHCs within insurance or payment-based systems. Eight primary healthcare clinics were recruited for the quantitative component based on a simple random sampling technique. The sample size of 380 was determined using standard statistical methods for population proportion. Initially, the unadjusted calculation yielded a required sample size of 384 for an infinite population. This was then adjusted for the finite population of 13,267 healthcare professionals in Almaty, resulting in a minimum required sample size of 377. To ensure robust representation, a final target of 380 respondents was set, maintaining a 95% confidence level and a 2.5% margin of error.

## Study design and data collection

A mixed-methods approach was used in this study to construct a better understanding of HIV-related stigma in primary healthcare. Specifically, we employed a sequential explanatory design, in which quantitative survey data were collected and analyzed first as the primary data source. Subsequently, qualitative in-depth interviews were conducted to complement and explain the survey findings. Lastly, we combined the findings from both phases in order to draw conclusions, focusing on how the qualiative data could better explain our quantative results. This design is illustrated in Fig 1.

The data collection process for this study involved contacting chief medical officers of eight randomly selected PHCs in Almaty to recruit healthcare providers for a cross-sectional survey, with further details provided in our earlier study [20]. The research team initially reached out to the chief medical officers of eight randomly selected PHCs in Almaty out of a total of 65 available PHCs for recruitment purposes. Following this, employees of these PHCs were invited to participate in a survey. Participation was voluntary, and the polyclinics' administrations were not informed about who chose to participate or not. Eligible respondents were individuals aged 18 or above, proficient in Kazakh and/or Russian languages, and with at least one year of work experience in healthcare. In total, 448 healthcare providers, including both clinical and non-clinical staff, participated in the study.

For the qualitative phase, 10 participants were selected from the survey sample for in-depth interviews. Using a purposive sampling strategy, we selected a mix of clinical and non-clinical personnel, including nurses, doctors, psychologists, and social workers. During the quantitative survey, participants were asked if they were willing to be contacted for a potential follow-up in-depth interview. Respondents who agreed to this provided a phone number for further contact. All identifying information such as full names or addresses was not collected to ensure confidentiality. From this pool of respondents who had consented to be re-contacted, 10 participants representing diverse clinical and non-clinical roles were purposively selected and invited to participate in one-on-one interviews at a time convenient for them.

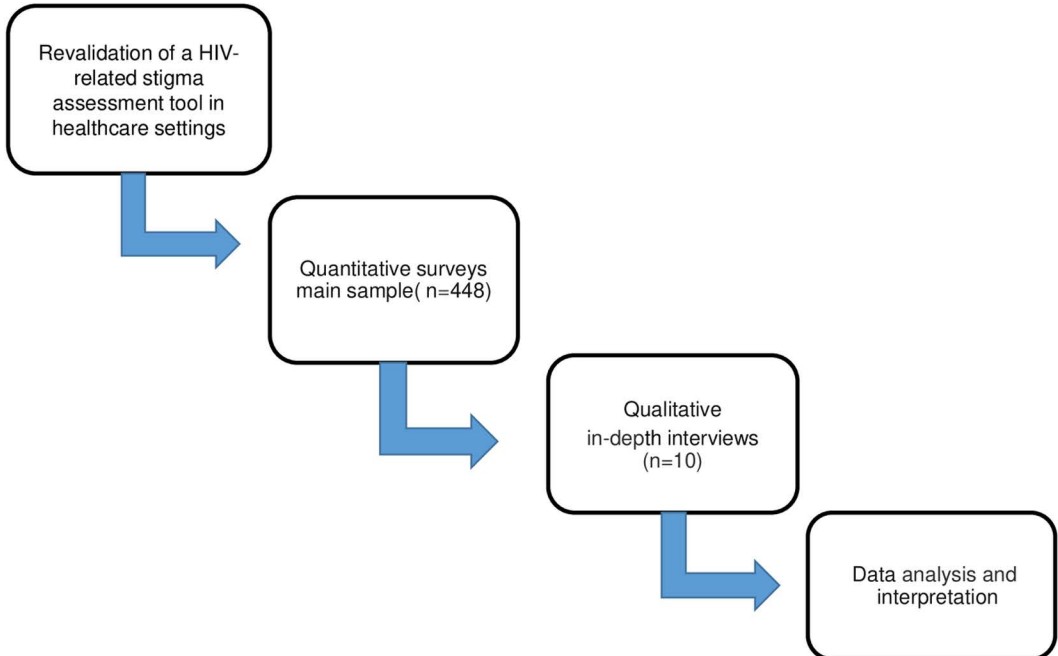

**Fig 1. Data collection process: Sequential integration of quantitative and qualitative methods.**

The cross-sectional surveys were conducted at the PHCs in Almaty from May 2, 2019 to July 2, 2019. Considering the workload of PHCs and the differences in work shifts among healthcare workers, we made the participation in surveys available throughout the day in the conference halls. Paper-based questionnaires were self-administered to provide more privacy to the respondents. A study investigator was available during the survey period to assist with any difficulties completing the surveys, and comprehensive instructions on the survey items were provided before respondents began filling out the questionnaires.

Due to COVID-19 restrictions, the qualitative interviews were conducted remotely in October 2021 using social media platforms such as WhatsApp, Zoom, and telephone calls. The interviews lasted between 30 and 50 minutes, and the interviewees were asked to specify the most convenient time and place where they could comfortably participate in the interviews. The interviews were audio recorded and transcribed later for the analysis.

## Measures

The study data were collected using anonymous self-report questionnaires for quantitative data and semi-structured interview guides for qualitative data. The original standardized assessment tool, validated in six countries (China, Dominica, Egypt, Kenya, Puerto Rico, and St. Christopher & Nevis), was employed to measure stigma through respondents' opinions about PLHIV. The tool demonstrated strong reliability in the original validation, with a Cronbach's alpha of 0.78 [21]. For the current study, we revalidated the tool in Kazakh and Russian, achieving excellent internal consistency, with a Cronbach's alpha of 0.86, indicating its high reliability for the study context. The revalidation process, including translation and administration in both languages, is comprehensively outlined in our previously published manuscript on the revalidation of a standardized HIV-related stigma assessment tool in healthcare settings in Almaty [20].

The questionnaire had several sections that included socio-demographic information, opinions about PLHIV, and fear of contracting HIV. Socio-demographic variables in this study included age, gender, professional position (e.g., clinical-nurse, surgeon, etc., and non-clinical-administrative staff, social workers, and other), ethnicity, religious affiliation (including self-reported religiousness), years of work in healthcare, and experience of working with PLHIV (e.g., "Among your patients in the past 12 months, did you have any patients who you knew to be HIV-positive?"). We also assessed ever-receiving training on HIV-related stigma topics (e.g., "Did you ever receive training in HIV-related stigma and discrimination"?) and knowledge of HIV transmission (correct answers to the provided HIV-transmission routes).

The primary study outcome was based on "Stigmatizing opinions about PLHIV," measured using a 4-point Likert scale with response options ranging from "Strongly Disagree" to "Strongly Agree". The number of people who agreed with at least one of the three stigmatizing statements such as "Most people living with HIV do not care if they infect other people", "People living with HIV should feel ashamed of themselves", "People get infected with HIV because they engage in irresponsible behaviors" and disagreed with the fourth statement "Women living with HIV should be allowed to have babies if they wish" were considered to have some level of stigmatizing opinion about PLHIV [22]. These responses served as a numerator for the stigmatizing opinion calculation, while the denominator was based on the number of healthcare staff who answered at least one of these statements [22]. Subsequently, a new binary outcome variable was created for further analysis including those who had some level of stigmatizing opinion about PLHIV and those who did not have it. In addition, we examined the willingness to provide medical services to HIV key populations with further clarifications on reasons that can affect it (e.g., I agree with the above-mentioned statement because: a. They put me at higher risk for disease).

Fear of HIV items included questions that were focused on fears of contracting HIV during basic medical interventions. Sample items on this scale included "How worried would you be about getting HIV if you did the following: 1. Touched the clothing of a patient living with HIV?; 2. Dressed the wounds of a patient living with HIV?; 3.Drew blood from a patient living with HIV". Response options included a scale of answers from "not worried" to "very worried'. "Not applicable" option was available for those who generally do not engage in medical manipulations. Similar to "Opinions about PLHIV" items, those who reported worry during at least one of the given medical services were treated as a numerator, while the

denominator included all those who answered at least one of the fear items [23]. Not applicable options and those who had not answered any of the fear of HIV items were sent to missing.

For the qualitative data collection, a semi-structured interview guide was used, consisting of 20 open-ended questions and guided probes with clarifications used throughout interviews (see S1 File. Qualitative interview guide.) The questions for the interviews were based on preliminary findings of the quantitative part, focusing on personal experiences with patients with HIV and opinions about PLHIV including key affected populations. Sample questions included, "What comes to your mind first when you think of people living with HIV?", "What do you think of people who get infected with HIV via sexual intercourse?", and "What is your personal attitude towards non-traditional sexual orientations and activities related to that topic? (worldwide and locally)".

### Data analysis

In the first stage of our analyses, we computed descriptive statistics using means and standard deviations (SD) for continuous variables and frequencies and percentages for categorical variables. Then raw bivariate associations and adjusted associations using multivariable logistic regression models, which were also utilized to explain variations, were employed to examine the links between socio-demographic factors and HIV-related stigma among the respondents. The selection of socioeconomic variables included in the logistic regression models was guided by the existing literature [13–15]. Fear of HIV items were not included in the regression model due to high numbers of not applicable responses. We identified these responses as missing following the questionnaire guide, which led to a decreased sample size to 272 complete responses. Our descriptive statistics, however, were based on all available data. Statistical significance was determined based on the p-value that is less than 0.05 and the corresponding 95% confidence interval (95% CI). All statistical analyses were carried out using R (v4.3.1; R Core Team 2021).

Qualitative interview data of the 10 participants were transcribed verbatim in the original language (Kazakh, Russian). Manual thematic analysis was used due to the bilingual nature of the interviews and their translation challenges. The interviews were coded following the section "Opinions about PLHIV" first deductively then depending on new findings, inductive codes were added. We used open and axial coding by categorizing the data in each section then we searched for repeated themes. The final version of the codebook consisted of three coding themes and eight sub-themes, which the analysts then applied across all the interviews. Qualitative data were analyzed in the original languages (Russian and Kazakh), and the primary analyst (BI), who is fluent in all three languages, translated participants' quotations into English. Other research team members (ZS and EK) with fluency in the respective languages reviewed and verified the translated quotes.

### Ethical considerations

Ethical approval for this research was obtained from Kazakh National Medical University Ethics Committee (IRB session №5/82). Ethical approval for research is granted for a period of one year in this institution. Since our study required the collection of qualitative data over an extended timeframe, we sought and obtained renewed ethical approval prior to conducting the in-depth interviews (IRB session №1/107). All the study participants were fully informed about the aims and methods of the research and about the choice of participating or not participating in the study. Written informed consent was obtained from the respondents before participating in both quantitative and qualitative arms of the study.

## Results

### Sociodemographic characteristics

A total of 448 healthcare staff were included in this study. Demographic characteristics of the sample are presented in Table 1. Females were the majority in this study sample (92%, n = 413) and nursing was the most common profession

**Table 1. Sociodemographic characteristics of the survey sample (n = 448).**

| Variables | Categories | N (%) |
|---|---|---|
| Gender | Male | 35 (8%) |
| | Female | 413 (92%) |
| Age group | 18–29 | 138 (31%) |
| | 30–40 | 78(17%) |
| | 41–51 | 97 (22%) |
| | >52 | 135 (30%) |
| Religion | Christian | 36 (8%) |
| | Islam | 366 (82%) |
| | Judaism | 3 (1%) |
| | Not religious | 26 (6%) |
| 7 (1%) is missing | Other | 10 (2%) |
| Ethnicity | Kazakh | 359 (80%) |
| | Russian | 34 (8%) |
| | Uighur | 20 (5%) |
| | Ukrainian | 6 (1%) |
| 7 (1%) is missing | Other | 22 (5%) |
| Professional category | Doctors/Physician | 99 (22%) |
| | Dentist | 16 (3.5%) |
| | Nurse | 274 (62%) |
| | Psychologist/Social worker | 19 (4%) |
| | Cleaning staff | 19 (4%) |
| 7 (1%) is missing | Other | 14 (3%) |

category (62%, n = 274). Such gender distribution among the respondents is not surprising since the majority of healthcare workers in the country is predominantly female. The descriptive statistics show the age of the respondents ranged from 19 to 74 (M = 40.02, SD = 13.92). The majority of the sample was ethnically Kazakh (81%, n = 359) and self-identified Muslim 83% (n = 366). Only 18% (n = 79) of the sample reported receiving training on HIV-related stigma and discrimination and even lesser 14% (n = 63) received training on discrimination towards key affected populations. The de-identified study dataset is provided as supporting information (see S2 Data. De-identified study dataset.)

## Quantitative results

Fig 2 demonstrates descriptive statistics of HIV-related stigma variables. Level of negative opinions towards patients with HIV was considerably high in this study 87% (n = 380) of the respondents agreed with at least one of the stigmatizing statements of the stigma scale. Similarly, 85% (n = 286 out of 335 involved in medical procedures) had some level of fear of HIV infection while providing medical care to PLHIV. Roughly one third of the respondents correctly answered all the HIV transmission questions (30%, n = 129), and around half of the respondents were aware about the undetectable viral load 53% (n = 212).

Unwillingness to provide medical care to the key affected populations of HIV was high in this study (Fig 3). Half of the respondents preferred not to provide medical services, if they had a choice, to SWs (52%, n = 211), PWID (54%, n = 233) and MSM (52%, n = 250), and a small number of respondents skipped these questions (5.50% to 8.50%). The most common reasons indicated for such unwillingness was "being put at risk of getting infected" (43–57%). The other reasons provided included "immoral behavior" and not having a specific training in working with HIV key populations and percentages of agreeing with such statements varied from 35% to 45% among respondents.

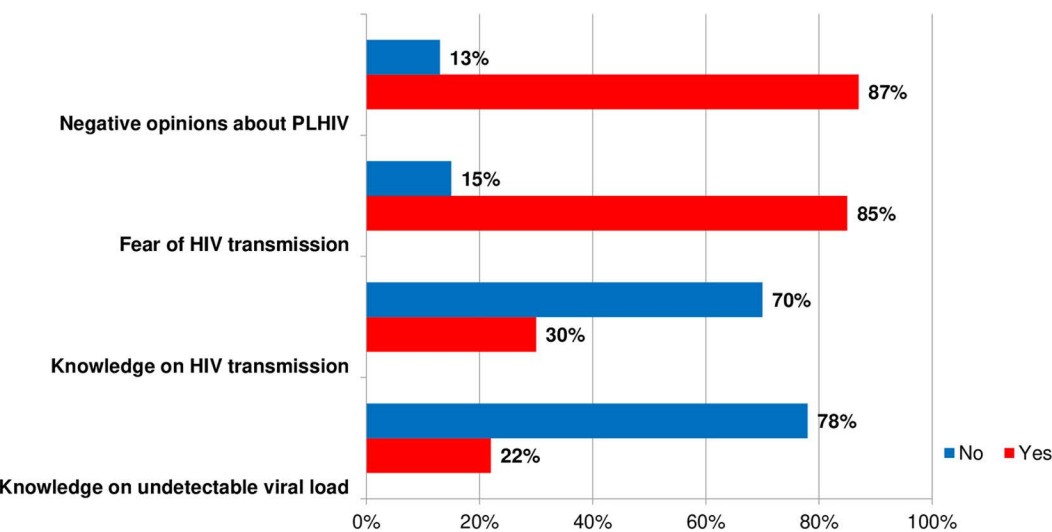

**Fig 2. Descriptive statistics of HIV-related stigma variables.**

*"If I had a choice, I would prefer not to provide services to:...."*

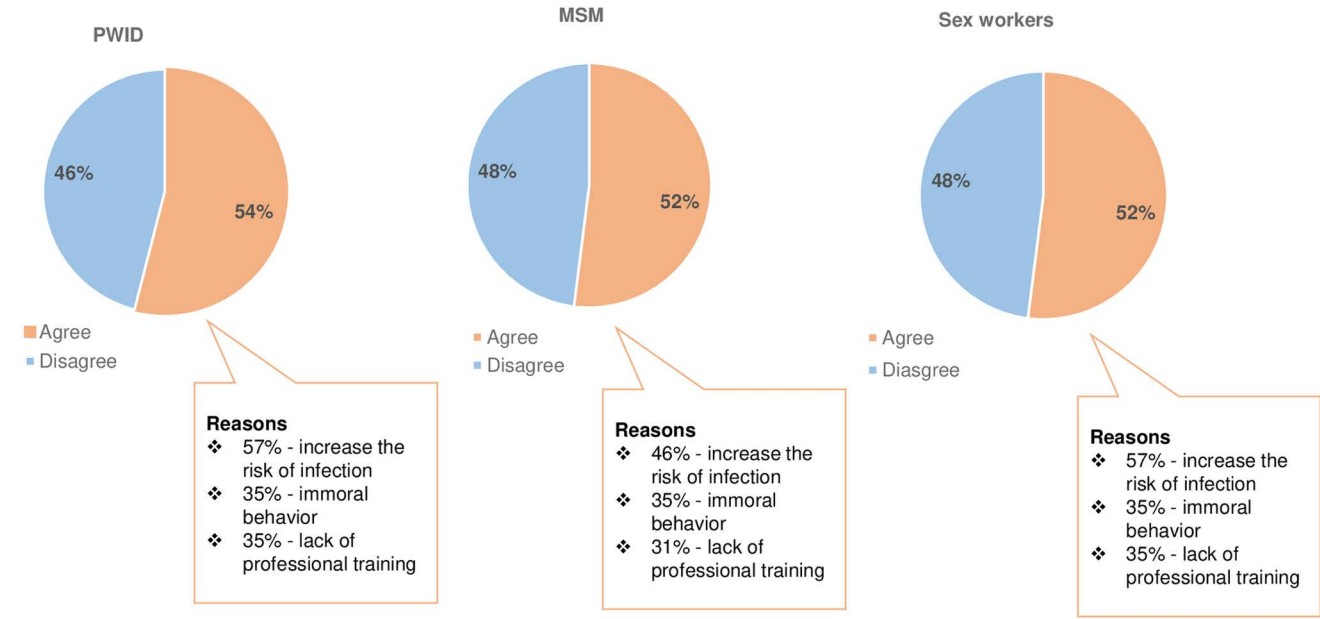

**Fig 3. Unwillingness to provide medical care to key affected populations if there was a choice.**

Table 2 demonstrates the crude and adjusted odds ratios (OR) obtained through logistic regression models. Multivariable logistic regression models,presented in Table 2, demonstrated a protective effect of longer years of work in healthcare on having stigmatizing opinions about PLHIV (AOR = 0.25; 95%CI = 0.09, 0.67; p = 0.006) while those who had not seen a patient living with HIV within the last 12 months had higher odds of holding stigmatizing opinions about PLHIV (AOR = 3.31; 95% CI = 1.73, 6.35; p = 0.001)

## Qualitative results

All qualitative interview participants were women (n = 10). The participants' ages ranged from 23 to 60 years, as presented in Table 3. Notably, only two of the ten participants reported having prior experience working with PLHIV.

Analysis of the qualitative data revealed that we were able to reach data saturation around the key findings from the quantitative data. The list of themes on opinions about PLHIV included fears over HIV transmission, empathetic feelings towards PLHIV, and negative feelings towards SWs and people with non-traditional sexual orientations (PWNSO) (Fig 4).

## Fear of getting infected with HIV at work

All of the participants reported adequate numbers of medical equipment to protect from HIV and protocols regarding patients with HIV. Almost all participants wished to have more training on the topic of HIV and proposed more interactive methods of training. Some participants also highlighted that they rarely see a patient with HIV on a daily basis and may forget many of its aspects.

Six out of ten interviewees mentioned that having a fear of HIV infection during medical care or treatment with patients with HIV is natural. However, due to this fear, some highlighted the need to perform extra precautions with PLHIV.

*"If, god forbid, we have such a patient, of course, we will still have to work with them, and we won't refuse to do so. But you know, with such a patient, we'll need extra precautions, of course, the approach will be different there." Nurse, 45 years old.*

## Empathy towards PLHIV and concerns

The qualitative interview participants frequently expressed concerns about the transmission of HIV when they thought of PLHIV, emphasizing the importance of preventive measures.

*"All I think is just the wish that they do not transmit it further." Pharmacist, 55 years old.*

**Table 2. Logistic regression results on determinants of HIV-related stigma.**

| Categorical variables | | | Negative opinions about PLHIV | | | |
|---|---|---|---|---|---|---|
| | | Frequency | Bivariate unadjusted OR [95%CI]. | p value | Multivariate adjusted OR [95%CI]. | p value |
| Age | | 40.02(13.92) | 0.98 (0.96,1) | 0.07 | 0.96 (0.93,1) | 0.08 |
| Position | Clinical | 410 (92) | ref | | ref | |
| | No clinical | 38(8) | 0.57 (0.23,1.61) | 0.27 | 0.59 (0.22,1.81) | 0.33 |
| Years of work in healthcare | >15 years | 210(45) | ref | | ref | |
| | 5-15 years | 99(24) | 0.50 (0.26,0.94) | 0.03 | 0.25 (0.09,0.67) | 0.006 |
| | <5 years | 139(31) | 1.75 (0.82,3.97) | 0.15 | 0.79 (0.20,3.22) | 0.74 |
| Religiousness | Not religious | 88(20) | ref | | ref | |
| | Slightly religious | 149(33) | 0.45 (0.18,1.09) | 0.06 | 0.59 (0.21,1.47) | 0.27 |
| | Moderately religious | 165(37) | 0.66 (0.26,1.66) | 0.26 | 0.60 (0.21,1.55) | 0.31 |
| | Strongly religious | 27(6) | 0.72 (0.17,3.03) | 0.64 | 0.65 (0.15, 3.39) | 0.57 |
| | Missing | 19(4) | | | | |
| Seen patients with HIV | Yes | 106(25) | ref | | ref | |
| | No | 323(75) | 2.88 (1.60,5.17) | < 0.001 | 3.31 (1.73,6.35) | < 0.001 |
| Knowledge of HIV transmission | Yes | 129(30) | ref | | ref | |
| | No | 292 (70) | 0.75 (0.42,1.38) | 0.35 | 0.83 (0.42,1.68) | 0.60 |

**Table 3. Demographic data of interview participants.**

| N | Age | Specialty | Years in healthcare | Experience of working with PLHIV |
|---|-----|-----------|---------------------|----------------------------------|
| 1 | 29 | Endocrinologist | 4 | No |
| 2 | 60 | Midwife | 35 | No |
| 3 | 55 | Pharmacist | 15 | No |
| 4 | 33 | Psychologist | 9 | No |
| 5 | 26 | Nurse | 4 | Yes |
| 6 | 43 | Pediatric nurse | 15 | No |
| 7 | 45 | Nurse | 28 | Yes |
| 8 | 35 | Midwife | 10 | No |
| 9 | 29 | Pediatrician | 4 | Yes |
| 10 | 23 | Nurse | 3 | No |

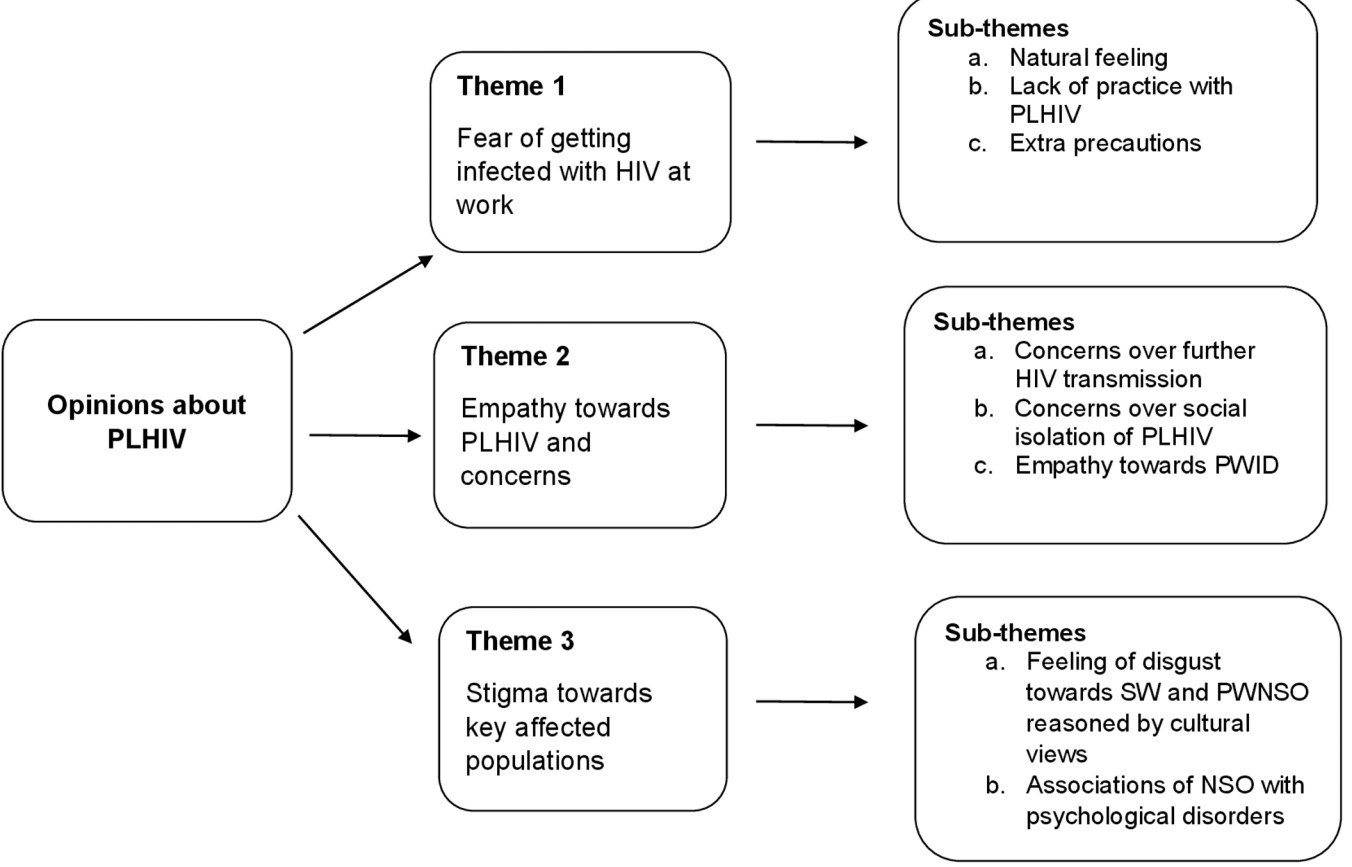

**Fig 4. Themes and sub-themes of the qualitative data.**

Additionally, many participants were aware of the psychosocial burden associated with HIV. They recognized the social isolation often experienced by PLHIV, reflecting a broader understanding of the dual challenges these individuals face: managing their health while also dealing with societal stigma.

*"I think such patients self-isolate themselves because of the judgment they may get due to their HIV diagnosis. Maybe their friends and relatives do not show enough support. I think that they do not want others to know about their HIV status, therefore they are forced to isolate themselves." Nurse, 26 years old.*

These concerns led to more questions about the possibility of PLHIV having families and children if they desired. None of the participants opposed the idea of PLHIV having children, but there was a lack of awareness or uncertainty about the risk of transmitting HIV to the fetus and the measures available to prevent perinatal HIV transmission.

Moving from general opinions about PLHIV towards specific key affected populations, interviews revealed more empathetic views about PWID. Seven out of ten participants acknowledged drug addiction as a condition that requires treatment. Some also considered them as victims of the illegal drug industry system.

*"But regarding drug users they are in a different category, first if all these people have a disorder. I mean they are addicted. They do not care if the needles are sterile or not, all they want is just to get their dose of the drug and that is it, nothing else bothers them..." Psychologist, 33 years old*

*"They are the victims of those who sell drugs for money. It is a profit to someone else. I feel bad for them." Midwife, 60 years old*

Additionally, the interviewees were more willing to offer help to PWID compared to other HIV key affected population groups.

*"I would offer help, and would say, "if you want to quit, let's work with psychiatrists and psychologists. If I can help anyhow, I definitely will."" Pediatrician, 29 years old*

## Stigma towards key affected populations

In addition to survey data, qualitative interviews revealed differential attitudes towards PLHIV based on the mode of HIV transmission (i.e. through blood transfusion or sexual transmission):

*"There are some patients who acquired HIV during medical procedures but regarding the drug users and prostitutes… I think they are infected because of their ignorance and lack of responsibility. It is their own fault, I think." Endocrinologist, 29 years old.*

Participants were also more inclined to the belief that HIV is spread mainly via "uncontrolled sexual behaviors" or from sex workers to men. According to the data, participants had more negative attitudes towards females, saying that men should be careful in choosing female sexual partners.

*"Most of the time, it is the men who get infected by visiting such promiscuous women. Those women, in my opinion, think that they should not be the only ones infected. If they have HIV, I think they want to infect others too." Nurse, 26 years old*

The majority of the participants of the qualitative interviews had negative attitudes towards sex workers, describing sex work as "disgusting", "immoral business", and feeling of "shame" if they knew anyone who was doing such a business.

*"In terms of that, I am against it, of course. Do not like it. I think it is a very dirty business since it is commercial. I guess there is nothing pure left about them since they are in that business, what else can I say." Nurse, 43 years old.*

*"I think those people are not ok psychologically. They need a lot of work on their psychology, you know, to turn them to the right path." Nurse, 26 years old.*

Similarly, negative opinions were seen predominantly towards PWNSO. They reported feelings of "discomfort" and "disgust" leading to unwillingness to be in contact with such individuals.

*"Thank God, I do not see such people, do not hear about them, and try to avoid them." Midwife, 60 years old.*

*"Oh, I feel really bad about it. I do not understand such people, and I do not wish that to anyone. Any person, if he has some kind of mind in his head, he will not do such things. I don't even want to think about it. But it also depends on society; if they have a good society, then it will not happen. If they have some kind of normal goal, then of course they can improve in life." Pediatric nurse, 43 years old.*

Furthermore, some participants suggested that non-traditional sexual orientation is culturally inappropriate for Kazakh society and that revealing one's sexual orientation may lead to serious consequences that people should care about.

*"I have negative attitudes towards gay people. Not sure about other countries, but in Kazakhstan it is weird to discuss such topics. For example, these days, if such people appear on social media platforms, then Kazakh men want to go find them and beat them up or something else. I have similar feelings like these men. It is just that they need to understand that promoting such things in Kazakhstan is stupid. We do not like people like that. I think they should think about it and about what it can lead to." Nurse, 26 years old*

The majority of the participants associated non-traditional sexual orientation with some kind of "psychological disorder" that also required immediate intervention.

*"I mean I am just shocked about them. But I think it is a psychological disorder. I think they have some psychological abnormalities." Nurse, 45 years old*

Some participants mentioned the difficulties of getting the medical care needed among PLHIV, especially in state clinics and primary healthcare centers. For example, there are trust rooms or social workers who are supposed to provide care for PLHIV in primary healthcare centers. However, such trust rooms were transformed into quarantine rooms in some settings during the emergence of the COVID-19 pandemic and were not functioning.

## Discussion

The aim of this study was to assess the level of HIV-related stigma in primary healthcare settings and the reasons leading to stigma. The findings suggested a notably high level of negative attitudes towards PLHIV in this sample. For example, only a quarter of the respondents did not agree with any of the stigmatizing statements about PLHIV in the questionnaire. Those who reported experience working with PLHIV within the last 12 months and those with longer years of work in healthcare in general were less likely to hold stigmatizing attitudes towards PLHIV.

These findings were further explored in the qualitative component of our study, which confirmed and contextualized these patterns. While the survey revealed widespread endorsement of negative beliefs toward PLHIV and key affected populations, the interviews clarified the moral and cultural judgments underlying these views. Although many participants expressed empathy for the psychosocial challenges faced by PLHIV, this empathy was often conditional and did not extend to those perceived as leading "immoral" lifestyles. Sex workers and MSM were frequently described in explicitly stigmatizing terms, suggesting that stigma is driven not only by fear of infection but also by entrenched moral biases.

Furthermore, fear of occupational exposure, reported by the majority of survey participants, was a recurring theme in interviews. Despite acknowledging the availability of protective measures, many participants described this fear as "natural" and emphasized the need to take extra precautions when treating PLHIV. This reflects an underlying anxiety that may contribute to differential or discriminatory clinical practices. The interviews also revealed limited direct contact with PLHIV, and that participants expressed a willingness to engage in more interactive and experience-based training to address HIV-related stigma and discrimination.

A particularly concerning finding was the differential treatment of PLHIV based on perceived infection mode. Stigma towards individuals engaged in same-sex behavior or drug use appeared to be stronger than towards PLHIV overall. Additionally, nearly all participants expressed negative opinions towards people with non-traditional sexual orientations and sex workers. These negative attitudes even extended to supporting physical violence against individuals perceived as "queers." Moreover, during the focus group discussions on the adaptation of the stigma assessment tool, there was a notable lack of awareness regarding MSM, prompting the research team to clarify study terms [20]. This phenomenon can be attributed to the absence of an open MSM community or discussions about sexual orientation in Kazakhstan, largely due to widespread homophobia in the country [24].

Numerous studies have explored HIV-related stigma in healthcare, primarily focusing on the experiences of PLHIV [6,15–18]. However, fewer studies have systematically investigated this issue among healthcare workers, including specific clinic staff such as physicians, nurses, and office managers. Despite this, existing literature consistently indicates moderate to high levels of stigma towards PLHIV within healthcare settings [13,17,18]. For instance, a study conducted among healthcare providers in Alabama and Mississippi of the United States (US) found comparable percentages of stigma, with 89% of respondents at urban healthcare centers and 91% at rural clinics demonstrating at least one stigmatizing attitude [25]. Similar findings have been observed in other countries such as Nigeria, China, Poland, and Iran [26–31].

The HIV Stigma Index surveys conducted among PLHIV in Kazakhstan in 2015 and 2020 consistently indicated that healthcare centers exhibited the highest levels of stigma and discrimination compared to other settings [6,18]. The main manifestations of stigma reported in the Stigma Index 2020 included disclosure of one's HIV status without consent, recommendations on not having sex, gossip among medical workers, and avoidance [18]. Similar to our findings, higher levels of experienced stigma and denials in care provision were seen among MSM and sex workers compared to other key affected populations (8.1% of cases among MSM and 23.3% among SWs). These specific groups also were the most frequently advised not to be involved in sexual relationships, verbally hurt, and were subjected to physical violence.

One of the explanations for such high rates of stigma can be the difference between specialized hospitals, such as AIDS centers, and non-specialized medical centers, such as primary healthcare settings. It is possible that medical workers who work in the latter may occasionally see HIV patients and have less experience in dealing with HIV daily. This has also been the case in our study, since seeing a patient with HIV had a protective effect on stigma. Additionally, some studies suggest healthcare providers in primary healthcare settings give differential treatment and disclose one's HIV status to take protective measures when dealing with HIV [15,28]. Nevertheless, these actions are considered a privacy violation of PLHIV, and clear guidance on unintentional discrimination is needed in future studies.

Widespread homophobia and lack of awareness likely contribute to the heightened HIV-related stigma observed in our study, particularly the strong negative attitudes towards MSM, sex workers, and individuals perceived as engaging in same-sex behavior. Despite the decriminalization of same-sex relationships, Kazakhstan still faces a pervasive climate of intense homophobia. According to one source, the local media often portray lesbian, gay, bisexual, transgender, queer, intersex, and asexual (LGBTQIA+) individuals in a negative light, leading to increased violence and hostility towards them in public spaces such as parks and nightclubs [24]. These vulnerable groups, including some from HIV communities, seem to experience the most discrimination in healthcare settings [6,18]. A 2022 study on human rights violations among LGBTQIA+ individuals, with and without HIV, found that Kazakhstan and Kyrgyzstan had some of the highest rates of

healthcare-related violations among EECA countries [32]. In particular, Kazakhstan reported the disclosure of HIV medical data in 10% of cases, a figure notably higher than the average (1.1%) observed across the countries studied. It is also important to note that research on attitudes towards LGBTQIA+ people, including our study, primarily focuses on major cities with higher levels of education. Therefore, it is plausible that other regions in the country may exhibit even higher rates of homophobia and stigma towards HIV and key populations.

Attitudes towards female sex workers in healthcare have been the subject of numerous studies, although there has been relatively less evidence from lower-income countries [33–35]. A systematic review across 50 countries with middle and low incomes found significantly higher rates of HIV among female sex workers compared to other women of reproductive age [36]. The review also found that healthcare settings often perpetuate the highest levels of enacted and perceived stigma. This phenomenon spans various cultures, with cultural conservatism in countries like Malaysia and Lebanon exacerbating taboos related to sex work or gender diversity within healthcare settings, affecting the willingness of both female and male sex workers to seek care. Similar challenges exist in Kazakhstan, where prostitution is legal in private settings except for brothels and pimping. Existing studies in the region suggest discrimination and violence against sex workers, including instances of threats or further abuse in response to reports of violence to the police [37,38].

The results of this study should be interpreted with several limitations in mind. First, as a cross-sectional study, we cannot establish causality for the associations found. Additionally, the study focused on PHCs in Almaty, limiting generalizability to other regions with high HIV prevalence in Kazakhstan, although findings are consistent with national studies on HIV stigma. It is also important to mention that although the estimated sample size for this study was 380, 448 eligible respondents were ultimately included. This was due to a high response rate within the data collection period. All eligible respondents were included to improve statistical power and representativeness. However, we acknowledge this as a deviation from the original protocol and recommend that future studies define stopping criteria in advance to manage such scenarios. Another limitation was the high number of "Not Applicable" responses in the "Fear of HIV-transmission at work" that limited the sample size and the decision of including this variable in the regression models. We discussed this limitation thoroughly in the earlier study [20]. Nonetheless, to our knowledge, this is the first study to address HIV-related stigma in healthcare using both quantitative and qualitative methods in Kazakhstan. The qualitative data provided a deeper understanding of HIV stigma in healthcare settings, particularly because the topic is sensitive and required further exploration. Another strength of the study is the validated survey instrument in both Kazakh and Russian languages, which enhances reliability. The inclusion of healthcare workers at all levels also provides a comprehensive view of stigma experiences within the healthcare system.

## Conclusion

Overall, our study reinforces existing research highlighting pervasive negative attitudes towards PLHIV among healthcare staff in primary healthcare settings. The additional value of this study lies in its finding that negative attitudes may not be limited to PLHIV specifically, but also extend to key affected populations, such as sex workers and men who have sex with men. While our research appears to align with the notion of "double stigma and double trouble" [39], this is an area to explore in further research in Kazakhstan. The observed association between reduced stigma and direct contact with patients with HIV may suggest the potential benefits of integrating HIV care into these settings to increase healthcare worker familiarity with PLHIV. Addressing the low levels of HIV knowledge among mid-level medical staff should be a priority in interventions targeting HIV-related stigma and discrimination. While acknowledging recent increases in training frequency since 2019, our study calls for replication studies across primary healthcare centers to track changes over time. We encourage these studies to focus on how stigma has changed over time and its influence on HIV management within the country. Effective strategies to combat HIV-related stigma should prioritize intervention quality over quantity of training sessions, drawing on international best practices that employ interactive methods such as group discussions, role-plays, and modular training covering stigma, infection control, and medical ethics. Additionally, in line with the socio-ecological

model, interventions should address multiple levels of stigma, including individual, family/relationship, community, and societal, while also considering intersectional stigma to create a more comprehensive approach [40].

## Supporting information

**S1 File. Qualitative interview guide.**
(DOCX)

**S2 Data. De-identified study dataset.**
(CSV)

## Acknowledgments

We express our sincere gratitude to all individuals and organizations whose contributions were essential to this research. We appreciate the support provided by the Fogarty International Center of the National Institutes of Health under Award Number D43TW010046 for offering biostatistics and epidemiology courses on HIV-related research during the first author's PhD program (Non- certificate degree at SUNY Albany). The content is solely the responsibility of the authors and does not necessarily represent the official views of the National Institutes of Health. We also deeply appreciate the study participants for their time and valuable input in completing the surveys. Special thanks to Sir Kanat Tosekbaev from the Department of Public Health of Almaty for his assistance in recruiting healthcare workers for the study. We sincerely thank the reviewers for their valuable time and insightful comments, which have helped strengthen our manuscript.

## Author contributions

**Conceptualization:** Balnur Iskakova, Elizabeth J. King.

**Data curation:** Balnur Iskakova, Zhamilya Nugmanova.

**Formal analysis:** Balnur Iskakova, Recai Murat Yucel.

**Investigation:** Balnur Iskakova, Recai Murat Yucel, Zhamilya Nugmanova.

**Methodology:** Elizabeth J. King, Recai Murat Yucel.

**Resources:** Jack DeHovitz.

**Software:** Elizabeth J. King, Jack DeHovitz.

**Supervision:** Jack DeHovitz, Zhamilya Nugmanova.

**Writing – original draft:** Balnur Iskakova, Zhamilya Nugmanova.

**Writing – review & editing:** Balnur Iskakova, Elizabeth J. King, Recai Murat Yucel, Jack DeHovitz, Zhamilya Nugmanova.

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
