## [Decision Letter · Decision Letter 0]

6 Nov 2024

PONE-D-24-31521"I think they are infected because of their ignorance and lack of responsibility": a mixed-methods study on HIV-related stigma in the healthcare system in Kazakhstan.PLOS ONE

Dear Dr. Iskakova,

Thank you for submitting your manuscript to PLOS ONE. After careful consideration, we feel that it has merit but does not fully meet PLOS ONE’s publication criteria as it currently stands. Therefore, we invite you to submit a revised version of the manuscript that addresses the points raised during the review process.

We look forward to receiving your revised manuscript.

Kind regards,

Joseph KB Matovu, Ph.D.

Academic Editor

PLOS ONE

**Journal Requirements:**

We express our sincere gratitude to all individuals and organizations whose contributions were essential to this research. Research reported in this publication was supported by the Fogarty International Center of the National Institutes of Health under Award Number D43TW010046. The content is solely the responsibility of the authors and does not necessarily represent the official views of   the National Institutes of Health. We also deeply appreciate the study participants for their time and valuable input in completing the surveys. Special thanks to Sir Kanat Tosekbaev from the Department of Public Health of Almaty for his assistance in recruiting healthcare workers for the study. We are thankful to Asfendiyarov Kazakh National Medical University and the New York State International Training and Research Program (D43 TW010046) for providing epidemiology and biostatistics training to our research team members. 

3. We note that this data set consists of interview transcripts. Can you please confirm that all participants gave consent for interview transcript to be published?

If they DID provide consent for these transcripts to be published, please also confirm that the transcripts do not contain any potentially identifying information (or let us know if the participants consented to having their personal details published and made publicly available). We consider the following details to be identifying information:

- Names, nicknames, and initials

- Age more specific than round numbers

- GPS coordinates, physical addresses, IP addresses, email addresses

- Information in small sample sizes (e.g. 40 students from X class in X year at X university)

- Specific dates (e.g. visit dates, interview dates)

- ID numbers

Or, if the participants DID NOT provide consent for these transcripts to be published:

- Provide a de-identified version of the data or excerpts of interview responses

- Provide information regarding how these transcripts can be accessed by researchers who meet the criteria for access to confidential data, including:

a) the grounds for restriction

b) the name of the ethics committee, Institutional Review Board, or third-party organization that is imposing sharing restrictions on the data

c) a non-author, institutional point of contact that is able to field data access queries, in the interest of maintaining long-term data accessibility.

d) Any relevant data set names, URLs, DOIs, etc. that an independent researcher would need in order to request your minimal data set.

For further information on sharing data that contains sensitive participant information, please see: https://journals.plos.org/plosone/s/data-availability#loc-human-research-participant-data-and-other-sensitive-data

If there are ethical, legal, or third-party restrictions upon your dataset, you must provide all of the following details (https://journals.plos.org/plosone/s/data-availability#loc-acceptable-data-access-restrictions):

a) A complete description of the dataset

b) The nature of the restrictions upon the data (ethical, legal, or owned by a third party) and the reasoning behind them

c) The full name of the body imposing the restrictions upon your dataset (ethics committee, institution, data access committee, etc)

d) If the data are owned by a third party, confirmation of whether the authors received any special privileges in accessing the data that other researchers would not have

e) Direct, non-author contact information (preferably email) for the body imposing the restrictions upon the data, to which data access requests can be sent.

Reviewers' comments:

Reviewer's Responses to Questions

**Comments to the Author**

1. Is the manuscript technically sound, and do the data support the conclusions?

Reviewer #1: Yes

Reviewer #2: Yes

2. Has the statistical analysis been performed appropriately and rigorously? 

Reviewer #1: Yes

Reviewer #2: I Don't Know

3. Have the authors made all data underlying the findings in their manuscript fully available?

Reviewer #1: Yes

Reviewer #2: No

4. Is the manuscript presented in an intelligible fashion and written in standard English?

Reviewer #1: Yes

Reviewer #2: Yes

5. Review Comments to the Author

**Reviewer #1: **Dear Authors,

Thank you for the opportunity to review your manuscript for publication. I appreciate the effort and dedication that went into this work. Please find the following comments for attention.

General comments:

• Insert page numbers throughout the manuscript

Abstract:

• (Line 32 under Methods) change “bivariate associations” into bivariate analysis to refer to the statistical analysis of the relationship between two variables.

• (Line 33 under methods) it would be better to say “models were performed” instead of “models were used” for a more precise and formal tone.

• (Line 40-42 under results) This statistical reporting “(AOR=0.25; 95%CI=0.09,0.67; p=0.006)”, must be formatted in full at first appearance to improve clarity as follows (AOR [Adjusted Odds Ratio] =0.25; 95%CI [95% Confidence Interval] =0.09, 0.6; p=0.006).

Introduction:

• (Line 60) Reformat and restate the following “…. estimates (2020)……” by simply removing parentheses as follows “…..estimates in 2020…”.

• (Line 62 -64) I recommend that the authors report on the achievement of the 2020 90-90-90 targets alongside the progress toward the new, ambitious 95-95-95 targets in the Kazakhstan context for viral suppression, set by UNAIDS for 2025.

• (Line 67-68) The following sentence must be cited, and I strongly recommend the citations of recent statistics “Notably, the number of new HIV infections in Kazakhstan increased by 68 39% in the years of 2010-2019.”

• (Line 74-76) This sentence warrants citation “The term ‘stigma’ is associated with “exclusion”, “rejection”, or “blame” of one due to holding a particular condition which is a subject for social judgment.”

• (Line 77-79) Citation is quired for this statement: “Stigma associated with HIV/AIDS can also be self-internalized by PLHIV, meaning that the general negative reactions to HIV from others may result in “internalized” stigmatization among PLHIV.”

• (Line 74-84) The multifaceted concept of stigma has been conceptualized, but this has primarily focused on self-stigma and enacted stigma. However, public stigma and perceived stigma, which may also be relevant in the context of this study, have been largely overlooked.

• (Line 82) Address the typo “feelings wile enacted stigma” write while for clarity

• (Line 83-84) This assertion needs citation “Fear of disclosure and shame caused by HIV-related stigma may keep PLHIV from getting the care needed in time.”

• (Line 87) Rephrase the term caregivers “For example, caregivers may” to healthcare providers.

• (Line 89) specify the context (country or region) in which this study was conducted “Previous research conducted in Europe…”

• (line 93-94) Authors ought to clarify what they mean with “regular procedure” given that this phrase carries significant ethical implications particularly with regards to patient confidentiality and ethical responsibility of Healthcare Providers. This point must be elaborated.

• (Line 107) Revise this grammatical error “exploratory study is aimed to….” to “This exploratory study aims to assess….”

Methods:

• (Line 109) This section should be written as “Material and Methods”

• (Line 116 -118) Under the subheading of “Population and Sampling”, it is essential to include the sample size formula along with its corresponding parameters defined to demonstrate how the total sample size of 380 was determined. This should clearly indicate whether the figure represents a minimum or maximum sample size. This transparency will enhance the understanding of the sampling process and ensure the validity of the research findings.

• (Line 120 – 124) Upon my review of this paragraph, it appears that the study utilizes a sequential mixed methods design; however, it does not specify which type of sequential design has been employed. It is important to note that there are two types of sequential mixed methods designs: exploratory and explanatory. Authors should first define the appropriate design relevant to the study and clarify its application within the research context.

• (Line 126) the following subheading should be well-formatted “Fig 1. Data collection steps”

• (Line 127 under data collection) This statement is very vague and therefore warrants clarification “The data collection process of this study is described elsewhere”

• (Line 140) Describe the structure of the self-administered survey questionnaire that was employed.

• (Line 150-151 under measures) Authors must state exactly where readers should look for the relevant information regarding re-validation process of the questionnaires.

• (Line 148 -183) The article should clearly describe the preexisting scales incorporated into the study for measuring key variables if the questionnaire was not originally developed by authors. It must report the documented reliability coefficients from previous studies and state the internal consistency coefficient obtained in the current study. Additionally, each measure’s items should be outlined by specifying that responses were rated on a Likert scale, including the number of points used for the ratings.

• (Line 184 -191) It is beneficial to include a figure that illustrates semi-structured interview guide questions.

• (Line 199) This statement must be cited “previous research findings.”

• (Line 200 -201) “We identified these responses as missing following 201 the questionnaire guide, which led to a decreased sample size to 272 complete responses” Authors should specify their approach to managing missing data, detailing the criteria or thresholds used to determine which incomplete questionnaires were excluded from the analysis. For example, questionnaires with more than 50% of missing data for each participant were excluded from the analysis.

• (Line 213-214) “Qualitative data analysis results were presented using quotations from participants and translated to English by the investigator who is fluent in all three languages.” Authors should clearly explain the translation process, detailing both back-and-forward translation methods to ensure conceptual equivalence. Additionally, they should discuss how the pilot study contributed to refining certain questions for improved clarity and context.

Results:

• (Line 228) Revise this phrase to “…was the most common…” instead of “the commonest..”

• (Line 230) Rephrase “…is homogeneous female dominated.” to “..is predominantly female”

• (Line 238) Remove colon by combining the sentence smoothly “study: 83% (n=380)”

• (Line 257) “[AOR=0.25, 95%CI (0.09,0.67), p=0.006)” Authors should be consistent between the use of parentheses and square bracket.

• (Line 277) Rephrase “medical manipulations” to “medical care or treatment”.

• (Line 278) The preferred and non-stigmatising term to refer to this key population is “patients/people living with HIV” instead of “HIV-positive patients”

Discussion:

• (Line 364) Results figures should be included exclusively in the results section and not in the discussion. (“with 87% (n = 380)”)

• (Line 368) Rephrase “HIV-positive patients” to PLHIV and please address this phrasing throughout the manuscript.

• (Line 371-372) Cite those studies immediately “Numerous studies have explored HIV-related stigma in healthcare, primarily focusing on the experiences of PLHIV.”

• (Line 406 -407) This line must be cited “…largely due to widespread homophobia in the country.”

• (Line 412) It is now LGBTQIA+ population and this acronym should be written in full at the first appearance.

• (Line 419) Correct the grammatical error “although there's been….” to “there has been…”

• (Line 432) Rephrase “other high HIV prevalence regions…” to “other regions with high HIV prevalence”

• (Line 434) Correct grammar “future research use larger,…” to “future research to use larger….”

• (Line 441-442) Authors must report reliability coefficients scores of all measures used to confirm reliability and the process of translation including forward-and back-translation (From English to Kazakh and Russian and from target languages back to the source language) should be also outlined for confirming content validity (Conceptual Equivalence).

Conclusions:

• (Line 446) This is now new “mid-level medical staff” whereas I initially believed the study was aimed at healthcare providers or professionals in general, without any specific focus on their level of training.

• (Line 448) Rephrase to remain consistent with “HIV care” instead of “AIDS care”

• (Line 453) Rephrase to “combat HIV related stigma” for consistency.

References:

• (Line 482) “STIGMA INDEX OF PEOPLE LIVING WITH HIV, 2.0. CAAPL- Central” revise formatting pertaining to the font size accordingly.

Thank you!!

**Reviewer #2: **Overall

Quantitative data were collected in 2019 (five years ago). It is likely that the HIV epidemic has changed. For example, most countries are nearing epidemic control. Is it possible that the findings are dated? If so, this could be included as a limitation.

Overall, our study reinforces existing research – What exactly is the additional value of this study?

Some stigma studies have recommended a socio-ecological model approach to addressing stigma focusing on the various levels (individual, family/relationship, community, societal). It is unlikely that just focusing on some health-care workers will solve HIV-related stigma. It is likely that in keeping with the socio-ecological model, interventions need to address multiple levels and address intersectional stigma – probably a discussion point. See for example: Ferris France N, Byrne E, Nyamwanza O, Munatsi V, Willis N, Conroy R, et al. Wakakosha "You are Worth it": reported impact of a community-based, peer-led HIV self-stigma intervention to improve self-worth and wellbeing among young people living with HIV in Zimbabwe. Front Public Health. 2023;11:1235150. Epub 20230728. doi: 10.3389/fpubh.2023.1235150

Abstract

Methods section should say when study was conducted.

Internalized stigma – Terminology is shifting to ‘internal stigma’.

Throughout, would put point (full stop after references i.e. [1,2].

Despite advances made in the clinical management of HIV, 24 % [24%]

do not have an access [do not have access]?

Methods

Interviews were conducted until data saturation was achieved. Was the sample of 10 predetermined or guided by the need to achieve theme saturation? It is unlikely that theme saturation could have been achieved with just 10 interviews.

The questionnaires were self-administered to provide more privacy to the respondents. Were these paper-based or electronic e.g. ACASI, CASI?

they can comfortably participate in the interviews [could]?

The questionnaire has several sections that include [had…included]

non-clinical-administrative stuff [staff]

people who agree with at least [agreed]

number of healthcare staff who answered at least of these statements [least ?? of]

who have some level of stigmatizing opinion about PLHIV and those who do not have it [who had…who did not]

Sample items on this scale include [included]

those who do not [did not]

those who have not answered [had not]

Line 159-160 and Table 2 knowledge on HIV transmission [of]?

For the qualitative data collection, a semi-structured interview guide was used consisting of 20 open-ended questions and guided probes with clarifications used throughout interviews. Please include as supplementary file.

reaching the word count of 15325 words – could leave out?

bilingual nature of the interviews and its [their]

Each interview was coded by a primary analyst and verified by a secondary analyst. Include initials if they are authors.

before the participation [before participating]

As shown in Table 3, the qualitative interview participants were women – There is nothing in Table 3 that shows they were women.

Separately, Table 3 could leave out the “Speciality” column as this likely identifies the participants e.g. One could guess who the young Endocrinologist is?

Analysis of the qualitative analysis??

The majority of the participants…See also Many interviewees mentioned, Most respondents – could include actual numbers. Also, these are only 10 participants who can’t be many even if all?

proposed on more [proposed more]?

These concerns have led [led]

drug addiction as a disease [condition]?

deep south USA [US]

during focus group discussions (FGDs) – It does not look like these are described anywhere else? Methods describe only qualitative interviews conducted remotely using social media platforms such as WhatsApp, Zoom and telephone calls

LGBT – provide full form first mentioned Terminology has shifted to LGBTQI+

It's important, it’s plausible [It is], there's been [there has]

Could present study strengths before limitations

future research use larger [uses]

6. PLOS authors have the option to publish the peer review history of their article (what does this mean?). If published, this will include your full peer review and any attached files.

Reviewer #1: No

Reviewer #2: No

---

## [Author Response · Author response to Decision Letter 1]

20 Dec 2024

Manuscript PONE-D-24-31521

Response to Reviewers

Dear Reviewers and the Editorial Team,

We would like to express our sincere gratitude for your time and effort in reviewing our manuscript titled "I think they are infected because of their ignorance and lack of responsibility": A Mixed-Methods Study on HIV-Related Stigma in the Healthcare System in Kazakhstan. Your thoughtful and thorough comments have been invaluable in improving the quality of our work.

We truly appreciate the detailed and constructive feedback you provided. Your insights have significantly contributed to enhancing the clarity of the manuscript. In response to your suggestions and comments, we have made the necessary revisions and ensured that all edits are clearly marked for your convenience. Specifically, we have used the review mode with track changes on the manuscript and highlighted any amendments made (in yellow). In the current document, we have provided our responses in blue font and included the new line numbers for your convenience.

Once again, thank you for your time, dedication, and expertise. We highly value your contribution to this work and look forward to any further comments you may have.

Sincerely,

Corresponding author,

Balnur Iskakova, PhD

School of Public Health

Asfendiyarov Kazakh National Medical University

Journal Requirements:

We have revised and corrected the manuscript to fully comply with Plos One's style requirements, including the specified file naming conventions.

Thank you for this comment. This research was conducted as part of the first author's PhD program at Asfedniyarov Kazakh National Medical University without specific funding. However, the first author benefited from training provided by the Fogarty International Center of the National Institutes of Health under Award Number D43TW010046 during her PhD studies. The acknowledgment section has been revised now as follows:

“We appreciate the support provided by the Fogarty International Center of the National Institutes of Health under Award Number D43TW010046 for offering biostatistics and epidemiology courses on HIV-related research during the first author’s PhD program (Non- certificate degree at SUNY Albany).”

3. We note that this data set consists of interview transcripts. Can you please confirm that all participants gave consent for interview transcript to be published?

If they DID provide consent for these transcripts to be published, please also confirm that the transcripts do not contain any potentially identifying information (or let us know if the participants consented to having their personal details published and made publicly available). We consider the following details to be identifying information:

- Names, nicknames, and initials

- Age more specific than round numbers

- GPS coordinates, physical addresses, IP addresses, email addresses

- Information in small sample sizes (e.g. 40 students from X class in X year at X university)

- Specific dates (e.g. visit dates, interview dates)

- ID numbers

Yes, we obtained explicit informed consent from the participants for the quotes from the transcripts to be published. This consent is documented and can be found in the interview guide, which we are attaching in three languages as part of our revised submission.

Regarding potentially identifying information, we ensured that the transcripts do not contain any such details. Specifically, we only collected participants' age in round numbers, information about their position or current role, years of work experience, and whether they have ever provided care to people living with HIV (PLHIV). No identifying information, such as names, specific ages, addresses, or other details listed in your guidelines, is included in the transcripts.

Comments to the Author

Reviewer #1

General comments:

• Insert page numbers throughout the manuscript

Thank you for your suggestion. We have now inserted page numbers throughout the manuscript, as requested.

Abstract:

• (Line 33/line 34 under Methods) change “bivariate associations” into bivariate analysis to refer to the statistical analysis of the relationship between two variables.

Thank you for your suggestion. We have made the requested change and replaced "bivariate associations" with "bivariate analysis" on line 34.

• (Line 34/line35 under methods) it would be better to say, “models were performed” instead of “models were used” for a more precise and formal tone.

Thank you for your suggestion. We have now made the requested change from “models were used” to “models were performed” on line 35.

• (Line 40-42/line 42-44 under results) This statistical reporting “(AOR=0.25; 95%CI=0.09,0.67; p=0.006)”, must be formatted in full at first appearance to improve clarity as follows (AOR [Adjusted Odds Ratio] =0.25; 95%CI [95% Confidence Interval] =0.09, 0.6; p=0.006).

Thank you for your suggestion. We have now made the requested change as follows:

“Logistic regression analysis indicated that longer years of work in healthcare were protective against stigmatizing opinions (Adjusted Odds Ratio (AOR) =0.25; 95% Confidence Interval (95%CI)=0.09,0.67; p=0.006), while not having seen an HIV-positive patient in the last 12 months was associated with higher stigma (AOR=3.31; 95%CI=1.73, 6.31; p<0.001)”

Introduction:

• (Line 61/line 61-64) Reformat and restate the following “…. estimates (2020)……” by simply removing parentheses as follows “…..estimates in 2020…”.

Thank you for the suggestion. We have revised and updated the prevalence statistics to reflect the most current data on line 61-65:

“There were 40,000 PLHIV in Kazakhstan by the latest estimates in 2023, and the highest prevalence of the infection is documented in several regions including Pavlodar, Karaganda, Almaty, North Kazakhstan, Kostanay, and East Kazakhstan [1,4,5].”

• (Line 65 -68/line 66-69) I recommend that the authors report on the achievement of the 2020 90-90-90 targets alongside the progress toward the new, ambitious 95-95-95 targets in the Kazakhstan context for viral suppression, set by UNAIDS for 2025.

Thank you for the recommendation. We have now included both the achievement of the 2020 90-90-90 targets and progress toward the 95-95-95 targets for viral suppression in Kazakhstan, as recommended on line 66-69.

“Although Kazakhstan has made progress, it has not yet achieved the 90-90-90 targets. Thus, in order for the country to reach the more ambitious 95-95-95 goals by 2025, substantial efforts are required, particularly in addressing stigma and discrimination”

• (Line 67-68/line 72-76) The following sentence must be cited, and I strongly recommend the citations of recent statistics “Notably, the number of new HIV infections in Kazakhstan increased by 68 39% in the years of 2010-2019.”

Thank you for your recommendation. We have now cited the relevant source and updated the prevalence statistics with the latest data. Additionally, we have revised the reference accordingly.

“Notably, HIV incidence in Kazakhstan surged by 88% from 2010 to 2021, making it the seventh highest increase globally, while the number of people living with HIV more than doubled [7]. The prevalence rates among key affected populations such as people who inject drugs (PWID), men who have sex with men (MSM), prisoners, and sex workers (SWs) were 7.60%, 8.80%, 4.20%, and 1.50%, respectively in 2023 [8].”

• (Line 78-79/line 80-82) This sentence warrants citation “The term ‘stigma’ is associated with “exclusion”, “rejection”, or “blame” of one due to holding a particular condition which is a subject for social judgment.”

Thank you for your recommendation. We have now cited the relevant source.

“HIV is one of the world’s highly stigmatized conditions. The term ‘stigma’ is associated with “exclusion”, “rejection”, or “blame” due to holding a particular condition, which is a subject for social judgment [9].”

9. Florom-Smith AL, De Santis JP. Exploring the concept of HIV-related stigma. Nursing Forum. 2012;47(3):153–65.

• (Line 77-79/line 82-85) Citation is required for this statement: “Stigma associated with HIV/AIDS can also be self-internalized by PLHIV, meaning that the general negative reactions to HIV from others may result in “internalized” stigmatization among PLHIV.”

Thank you for your recommendation. We have now cited the relevant source.

“Stigma associated with HIV/AIDS can also be internalized by PLHIV, meaning that the general negative reactions to HIV from others may result in “internalized” stigmatization among PLHIV [10].”

• (Line 74-84/line 79-90) The multifaceted concept of stigma has been conceptualized, but this has primarily focused on self-stigma and enacted stigma. However, public stigma and perceived stigma, which may be relevant in the context of this study, have been largely overlooked.

We appreciate the reviewer’s thoughtful comments on the multifaceted concept of stigma. While various terms are used to conceptualize stigma, in this study, we chose to focus specifically on anticipated, internalized, and enacted stigma. Our intention was to highlight the primary forms of internal and external stigma relevant to our findings without delving too deeply into the multiple subtypes. However, we recognize the importance of public and perceived stigma, and we note that perceived stigma closely aligns with anticipated stigma, while public stigma parallels the enacted stigma we addressed. We are open to incorporating a brief reflection on these aspects in the Discussion section. If the reviewer has specific resources or references to recommend, we would gladly consider including them to enrich our manuscript further.

• (Line 86/line 88) Address the typo “feelings wile enacted stigma” write while for clarity

Thank you for pointing out the typo errors. We have corrected and it now reads as “Internalized stigma may lead to low self-esteem, self-imposed withdrawal, and even suicidal feelings, while enacted stigma may exacerbate such feelings, resulting in an unwillingness to engage with others [10-12].”

• (Line 87-88/line 89-90) This assertion needs citation “Fear of disclosure and shame caused by HIV-related stigma may keep PLHIV from getting the care needed in time.”

Thank you for your recommendation. We have now cited the relevant source on line 88-89.

“Fear of disclosure and shame caused by HIV-related stigma has been shown to keep PLHIV from getting the care needed in time [11].”

11. Dessie ZG, Zewotir T. HIV-related stigma and associated factors: a systematic review and meta-analysis. Frontiers in Public Health. 2024 Jul 23;12:1356430. .

• (Line 87/line 93) Rephrase the term caregivers “For example, caregivers may” to healthcare providers.

Thank you for your suggestion. We have now rephrased “the caregivers” to “healthcare providers” on line 93.

• (Line 94/line 95-97) specify the context (country or region) in which this study was conducted “Previous research conducted in Europe…”

We realized that the cited European paper referenced international research. Therefore, we updated the reference and paraphrased the sentence in the manuscript. The revised sentence now reads:

“Studies on HIV-related stigma conducted in China and Vietnam suggest that healthcare staff may adopt specific attitudes and behaviors to align with their peers and gain social acceptance [13,14].”

• (line 93-94/line 99-101) Authors ought to clarify what they mean with “regular procedure” given that this phrase carries significant ethical implications particularly with regards to patient confidentiality and ethical responsibility of Healthcare Providers. This point must be elaborated.

We agree with this point and have revised the phrasing accordingly. The term “regular procedure” has been clarified to better reflect the ethical concerns surrounding patient confidentiality and the responsibility of healthcare providers.

“Although some healthcare providers in resource-limited settings may prefer to visibly mark the files PLHIV for perceived safety reasons, this practice can lead to breaches of patient confidentiality and reinforce stigma [14].”

• (Line 111/line 113) Revise this grammatical error “exploratory study is aimed to….” to “This exploratory study aims to assess….”

Thank you for your suggestion. We have now made the requested grammatical change and now it reads as “This exploratory study aimed to assess HIV-related stigma in Kazakhstani primary healthcare settings and the factors leading to stigmatization of PLHIV using a mixed methods approach”

Methods:

• (Line 109/line 116) This section should be written as “Material and Methods”

Thank you for your suggestion. We have now changed the heading to “Materials and methods”

• (Line 116 -118/line 123-129) Under the subheading of “Population and Sampling”, it is essential to include the sample size formula along with its corresponding parameters defined to demonstrate how the total sample size of 380 was determined. This should clearly indicate whether the figure represents a minimum or maximum sample size. This transparency will enhance the understanding of the sampling process and ensure the validity of the research findings.

Thank you for your comment. The sample size for this study was determined based on the most recent data, which indicated that approximately 13,267 healthcare professionals, including doctors and nursing staff, were employed in medical institutions across Almaty in 2017. The required sample size was calculated using a population proportion formula for infinite populations:

n0=E2Z2⋅P⋅(1−P)

2

This initial calculation yielded a sample size of n0=384. To account for the finite population size, the sample size was adjusted using the formula:

n adjusted=1+N(n0−1)n0

where N represents the total population. This adjustment resulted in a minimum required sample size of 377 respondents. For simplicity and statistical robustness, a final sample size of 380 was used, ensuring representativeness of the healthcare workforce in Almaty with a 95% confidence level and a 5% margin of error.

Based on this we have also modified the sample size information in the revised manuscript and it reads as:

“The sample size of 380 was determined using standard statistical methods for population proportion. Initially, the unadjusted calculation yielded a required sample size of 384 for an infinite population. This was then adjusted for the finite population of 13,267 healthcare professionals in Almaty, resulting in a minimum required sample size of 377. To ensure robust representation, a final target of 380 participants was set, maintaining a 95% confidence level and a 5% margin of error.”

• (Line 120 – 124/line 130-135) Upon my review of this paragraph, it appears that the study utilizes a sequential mixed methods design; however, it does not specify which type of sequential design has been employed. It is important to note that there are two types of sequential mixed methods designs: exploratory and explanatory. Authors should first define the appropriate design relevant to the study and clarify its application within the research context.

Thabk you for the comment. We used an explanatory sequential mixed methods design. The first phase involved a quantitative survey, followed by a qualitative phase to provide deeper insights that could explain and elaborate on the

---

## [Decision Letter · Decision Letter 1]

22 Jan 2025

PONE-D-24-31521R1"I think they are infected because of their ignorance and lack of responsibility": a mixed-methods study on HIV-related stigma in the healthcare system in Kazakhstan.PLOS ONE

Dear Dr. Iskakova,

Thank you for submitting your manuscript to PLOS ONE. After careful consideration, we feel that it has merit but does not fully meet PLOS ONE’s publication criteria as it currently stands. Therefore, we invite you to submit a revised version of the manuscript that addresses the points raised during the review process.

We look forward to receiving your revised manuscript.

Kind regards,

Joseph KB Matovu, Ph.D.

Academic Editor

PLOS ONE

Journal Requirements:

Additional Editor Comments:

This is an important study about stigma among primary healthcare workers. Although the data were collected some 4-5 years, I believe that the study findings are still relevant to the HIV response globally. I have a few comments in addition to those raised by the reviewers:

1. The authors indicate that for the qualitative component, they interviewed 10 participants were chosen "... from the main sample" for in-depth interviews. It is not clear how and when the 10 participants were selected. Were the 10 participants also interviewed quantitatively? If yes, did the selection happen immediately after the quantitative interview had been completed? If they were interviewed after the quantitative interview, were they informed that they would be invited for a follow-on qualitative interview? In general, the authors should explain: a) how and when the selection of the 10 participants was made, and b) why the selection was restricted to only 10 participants.

2. The authors use the term "participants" to refer to those who participated in the quantitative and qualitative interviews. This can be confusing. I suggest that they use 'respondents' for the quantitative interviews, and 'participants' for the qualitative interviews.

3. It wasn't clear to me if the 10 participants were key informants or in-depth interview participants. While the rest of the paper refers to "in-depth interviews" - on page 11, the authors refer to these participants as "informants". So, were they interviewed as key informants or in-depth interview participants?

4. The estimated sample size was 380 but the authors report that they interviewed 448 respondents. No explanation is given for this sort of protocol deviation. Did they seek an amendment to increase the sample size from 380 to 448? Why and how did they end up interviewing more than the estimated sample size? What was the rationale for this? The authors should provide a clear explanation/justification for going over their estimated sample size by up to 68 respondents.

5. Regarding the HIV stigma measurements, can the authors reference any standard tool that they used to collect data? I would like to imagine that stigma validated tools exist and there are standard questions that are used to measure stigmatizing attitudes about HIV, including those used in demographic and health surveys, stigma index, etc. Where did the authors obtain their questions to measure the stigmatizing opinions about HIV?

6. To what extent are the HIV stigma measurements consistent with other measures of HIV stigma in other settings/countries?

Reviewers' comments:

Reviewer's Responses to Questions

**Comments to the Author**

1. If the authors have adequately addressed your comments raised in a previous round of review and you feel that this manuscript is now acceptable for publication, you may indicate that here to bypass the “Comments to the Author” section, enter your conflict of interest statement in the “Confidential to Editor” section, and submit your "Accept" recommendation.

Reviewer #1: All comments have been addressed

Reviewer #2: (No Response)

2. Is the manuscript technically sound, and do the data support the conclusions?

Reviewer #1: Yes

Reviewer #2: (No Response)

3. Has the statistical analysis been performed appropriately and rigorously? 

Reviewer #1: Yes

Reviewer #2: (No Response)

4. Have the authors made all data underlying the findings in their manuscript fully available?

Reviewer #1: Yes

Reviewer #2: (No Response)

5. Is the manuscript presented in an intelligible fashion and written in standard English?

Reviewer #1: Yes

Reviewer #2: (No Response)

6. Review Comments to the Author

Reviewer #1: Discussion:

Comment 1:

(Line 394-395) “For instance, a study conducted among healthcare providers in the Deep South US found comparable percentages of stigma, with 89% of respondents at urban healthcare centers and 91% at rural clinics demonstrating at least one stigmatizing attitude [25].”

Specify the context in which the study was conducted. Change “Deep South US” to “among healthcare staff in Alabama and Mississippi of the United States (US)”.

Comment 2:

(Line 387) Rephrase “stigmatizing opinions about PLHIV” to Stigmatizing attitudes towards PLHIV.

Comment 3:

Throughout the discussion section, it would be clearer and more reader-friendly if the either authors first state the key results or findings of the current study before delving into a comparison with previous studies. Presenting the results upfront ensures the readers understand the main contributions of the study before they evaluate how these findings relate to existing literature. This approach helps to avoids confusion and strengthens the narrative flow of the discussion without missing the bigger picture. Paragraph from line 417 to 426 is an exception to the above comment.

Comment 4:

In the discussion paragraphs (from line 419 to 449), the authors addressed discriminatory and stigmatizing incidents against key populations, including MSM, LGBTQIA+, and sex workers. It is imperative to clarify whether these vulnerable groups are also part of the PLHIV community, experiencing compounded stigmatization and discrimination, particularly in primary healthcare settings when seeking HIV care. Otherwise, the stigma discussed may not be directly related to HIV as the primary focal point of the current study.

Reviewer #2: Most of my comments have been addressed. I only have a few more, outlined below.

• Stigma towards individuals engaged in same-sex behavior or drug use appeared to be stronger than towards PLHIV overall. Unsure how respondents were able to make the distinction. What if those in same-sex behavior or drug use were also LHIV? Just wondering if the issue of double-stigma emerged or should be explored in future studies – that is both same-sex behavior or drug use and LHIV. See for example: the “double stigma and double trouble” reported by https://doi.org/10.1371/journal.pgph.0002442

• …around 82% of PLHIV in the country are aware of their status, 68% are on ART, and only around 78% have suppressed viral loads [5].

These figures are confusing as one would expect a cascade - with first figure being highest, second being higher and third being lower than second.

• Although Kazakhstan has made progress, it has not yet achieved the 90-90-90 targets.

Include a citation of the 90-90-90 targets and also the more ambitious 95-95-95 goals by 2025.

• Although some healthcare providers in resource-limited settings may prefer to visibly mark the files PLHIV for perceived safety reasons, this practice can lead to breaches of patient confidentiality and reinforce stigma [15]. Include [of]: Although some healthcare providers in resource-limited settings may prefer to visibly mark the files of PLHIV for perceived safety reasons…

• were restricted to small sample size studies and/or focusing – change to [focused]

• leading to stigmatization of PLHIV using a mixed methods approach. Elsewhere, spelling includes hyphen: mixed-methods

• There are 65 PHCs in Almaty located evenly among 8 districts. 65/8 may not be evenly distributed?

• Several primary healthcare clinics (n=8) were recruited for the quantitative component based on a simple random sampling technique. Could say: Eight primary healthcare clinics were recruited for…

• where they can comfortably – change to [could] comfortably

• The primary study outcome was based "Stigmatizing opinions about PLHIV," measured using a 4-point Likert scale with response options ranging from “Strongly Disagree” to “Strongly Agree”.The – Is its based [on]? Also check space between sentences.

• For the qualitative data collection, a semi-structured interview guide was used consisting of 20 open-ended questions and guided probes with clarifications used throughout interviews (see S1 File) – include reference to the supplement file as suggested.

• which were also utilized to explain variations – see additional spaces

• Written informed consent forms were obtained from the respondents before participating in both quantitative and qualitative arms of the study - Written informed consent was obtained from the respondents before participating in both quantitative and qualitative arms of the study.

• Such gender distribution among the respondents is not surprising since the majority of healthcare workers in the country is predominantly female dominated – delete dominated: is predominantly female.

• 14%(n=63) – check space

• Fig 2 demonstrates descriptive statistics of HIV related stigma [HIV-related]

• around a half of the respondents were aware about the undetectable viral load [around half…were aware of…]

• with non-traditional sexual orientations or PWNSO (Fig 4) – give full form of PWNSO here.

• In other words, only one quarter… [For example, only a quarter…]

• one’s HIV status without a consent [one’s HIV status without consent]

• In our study, we observed differential treatment of PLHIV based on [perceived infection mode].

• According to one source, the local media often portrays s lesbian [delete extra “s” after portrays

• Nonetheless, [to our knowledge], this is the first study to address HIV-related stigma in healthcare using both quantitative and qualitative methods in Kazakhstan.

• Effective strategies to combat HIV related stigma [HIV-related]

• Acknowledgement [Acknowledgements]

7. PLOS authors have the option to publish the peer review history of their article (what does this mean?). If published, this will include your full peer review and any attached files.

Reviewer #1: No

Reviewer #2: No

---

## [Author Response · Author response to Decision Letter 2]

5 Mar 2025

Manuscript PONE-D-24-31521

Response to Reviewers

Dear Reviewers and the Editorial Team,

We would like to thank you for your time and effort in reviewing our manuscript titled "I think they are infected because of their ignorance and lack of responsibility": A Mixed-Methods Study on HIV-Related Stigma in the Healthcare System in Kazakhstan.

In response to your suggestions and comments, we have made the necessary revisions and ensured that all edits are clearly marked for your convenience. Specifically, we have used the review mode with track changes in the manuscript document and highlighted all amendments in yellow in the response letter. In the current document, we also provided our responses in blue font along with updated line numbers for ease of reference.

Once again, thank you for your time, dedication, and expertise. We highly value your contribution to this work and look forward to any further comments you may have.

Sincerely,

Corresponding author.

Journal Requirements:

Thank you for your feedback regarding the reference list. We have carefully reviewed and corrected the duplicated citations in our manuscript. Specifically:

1. Citation UNAIDS. Kazakhstan [Internet] was cited twice (as references 1 and 8). We have replaced the first citation with:

• Obeagu EI, Obeagu GU. A Review of knowledge, attitudes and socio-demographic factors associated with non-adherence to antiretroviral therapy among people living with HIV/AIDS. Int. J. Adv. Res. Biol. Sci. 2023;10(9):135-42.

2. Citation Davlidova S, Haley-Johnson Z, Nyhan K, Farooq A, Vermund SH, Ali S. Prevalence of HIV, HCV and HBV in Central Asia and the Caucasus: A systematic review was cited twice (as references 3 and 6). We have deleted the duplicate citation and replaced it with the corresponding source that was missing in our previous submission.

Additionally, we have verified that none of the references in our updated list includes retracted papers.

Additional Editor Comments:

This is an important study about stigma among primary healthcare workers. Although the data were collected some 4-5 years, I believe that the study findings are still relevant to the HIV response globally. I have a few comments in addition to those raised by the reviewers:

1. The authors indicate that for the qualitative component, they interviewed 10 participants were chosen "... from the main sample" for in-depth interviews. It is not clear how and when the 10 participants were selected. Were the 10 participants also interviewed quantitatively? If yes, did the selection happen immediately after the quantitative interview had been completed? If they were interviewed after the quantitative interview, were they informed that they would be invited for a follow-on qualitative interview? In general, the authors should explain: a) how and when the selection of the 10 participants was made, and b) why the selection was restricted to only 10 participants.

Thank you for your important comments. We have addressed your questions regarding the selection of participants for the qualitative component of the study as follows:

a) How and when the selection of the 10 participants was made?

All participants in the quantitative component were informed during the survey process that they might be invited to participate in a follow-up qualitative interview. To maintain confidentiality, we collected phone or work contact numbers from survey respondents for the interview phase. Using a purposive sampling strategy, the 10 participants selected for the qualitative interviews were chosen from these contacts, which included a mix of nurses, doctors, and non-clinical staff such as psychologists and social workers.

The cross-sectional surveys were conducted at the primary healthcare centres (PHCs) in Almaty from May 2, 2019, to July 2, 2019. The qualitative interviews, however, were conducted later, remotely, using platforms like WhatsApp, Zoom, and telephone calls due to COVID-19 restrictions in October 2021. This gap between quantitative and qualitative phases was also due to the primary investigator’s academic commitments abroad (non-certificate degree in the USA as a part of her PhD) during that period. After the investigator’s return to Almaty and after extending the ethical approval for the qualitative phase, the selected participants were invited for the follow-up interviews. We have clarified this timeline and the process in the manuscript.

Specifically, we have revised lines 146-150 and 156 to include this explanation, ensuring the process is clear:

• Line 146-148: "For the qualitative phase, 10 participants were selected from the survey sample for the in-depth interviews. Using a purposive sampling strategy, we selected a mix of clinical and non-clinical personnel, including nurses, doctors, psychologists, and social workers.”

• Line 149-150: "The cross-sectional surveys were conducted at the PHCs in Almaty from May 2, 2019, to July 2, 2019."

• Line 156-158: "Due to COVID-19 restrictions, the qualitative interviews were conducted remotely using social media platforms such as WhatsApp, Zoom, and telephone calls due to COVID-19 in October, 2021."

b) Why the selection was restricted to only 10 participants?

The selection of 10 participants was based on the principle of thematic saturation, as guided by the key findings from the quantitative data. The qualitative phase was intended to supplement the larger quantitative phase and therefore more limited in its scope. We have been sure to present themes for which we achieved data saturation among the 10 participant sample. We have specified this reasoning in the "Qualitative Results" section of the manuscript (lines 288-291), where we explain:

" Analysis of the qualitative data revealed that we were able to reach data saturation around the key findings from the quantitative data. The list of themes on opinions about PLHIV included fears over HIV transmission, empathetic feelings towards PLHIV and negative feelings towards SWs and people with non-traditional sexual orientations (PWNSO) (Fig 4).”

We hope this clarification resolves your concerns.

2. The authors use the term "participants" to refer to those who participated in the quantitative and qualitative interviews. This can be confusing. I suggest that they use 'respondents' for the quantitative interviews, and 'participants' for the qualitative interviews.

Thank you for your suggestion. We have followed your recommendation and made the necessary changes in the manuscript. We have now used "respondents" to refer to those who participated in the quantitative surveys and "participants" for the qualitative interviews to avoid any confusion.

3. It wasn't clear to me if the 10 participants were key informants or in-depth interview participants. While the rest of the paper refers to "in-depth interviews" - on page 11, the authors refer to these participants as "informants". So, were they interviewed as key informants or in-depth interview participants?

Thank you for pointing this out. To clarify, the 10 individuals were indeed qualitative interview participants, not key informants. We have revised the manuscript to consistently refer to them as "participants" throughout the manuscript, including on page 11, to avoid any confusion.

4. The estimated sample size was 380 but the authors report that they interviewed 448 respondents. No explanation is given for this sort of protocol deviation. Did they seek an amendment to increase the sample size from 380 to 448? Why and how did they end up interviewing more than the estimated sample size? What was the rationale for this? The authors should provide a clear explanation/justification for going over their estimated sample size by up to 68 respondents.

Thank you for pointing out this. Given the positive response to the invitations to the surveys, we decided to include all participants who met the eligibility criteria within the data collection timeframe. This decision was made to ensure inclusivity, maintain a representative sample, and utilize the additional data to enhance the robustness of our findings, i.e. higher statistical accuracy and power. Additionally, we wanted to allow for potential nonresponse, which could affect the representativeness of the sample. Allowing for larger sample size would also allow for higher statistical accuracy and power.

5. Regarding the HIV stigma measurements, can the authors reference any standard tool that they used to collect data? I would like to imagine that stigma validated tools exist and there are standard questions that are used to measure stigmatizing attitudes about HIV, including those used in demographic and health surveys, stigma index, etc. Where did the authors obtain their questions to measure the stigmatizing opinions about HIV?

Thank you for your question. The tool we used to assess HIV stigma is a brief, standardized assessment that has been validated across six countries (China, Dominica, Egypt, Kenya, Puerto Rico, and St. Christopher & Nevis). We referenced the following study for the original validation of this tool:

Nyblade L, Jain A, Benkirane M, Li L, Lohiniva AL, McLean R, Turan JM, Varas‐Díaz N, Cintrón‐Bou F, Guan J, Kwena Z. A brief, standardized tool for measuring HIV-related stigma among health facility staff: results of field-testing in China, Dominica, Egypt, Kenya, Puerto Rico, and St. Christopher & Nevis. Journal of the International AIDS Society. 2013 Nov;16:18718. [21]

Additionally, we revalidated the tool for use in Kazakhstan, specifically in Kazakh and Russian, through translation, factor analysis, and focus group discussions (FGD) in our earlier publication. This process was detailed in a previous publication, which we have cited as reference [20].

We have now specified that we used the standardised stigma assessment tool in the methods section (Lines 162-171):

“The study data were collected using anonymous self-report questionnaires for quantitative data and semi-structured interview guides for qualitative data. The original standardized assessment tool, validated in six countries (China, Dominica, Egypt, Kenya, Puerto Rico, and St. Christopher & Nevis), was employed to measure stigma through respondents’ opinions about PLHIV. The tool demonstrated strong reliability in the original validation, with a Cronbach's alpha of 0.78 [21]. For the current study, we revalidated the tool in Kazakh and Russian, achieving excellent internal consistency, with a Cronbach’s alpha of 0.86, indicating its high reliability for the study context. The revalidation process, including translation and administration in both languages, is comprehensively outlined in our previously published manuscript on the revalidation of a standardized HIV-related stigma assessment tool in healthcare settings in Almaty [20].”

"Stigmatizing opinions about PLHIV" was measured using the same standardized and re-validated tool on assessing stigma in healthcare based on 5 items which is explained in the following section if the manuscript. Lines 181-190:

“The primary study outcome was based on "Stigmatizing opinions about PLHIV," measured using a 4-point Likert scale with response options ranging from “Strongly Disagree” to “Strongly Agree”. The number of people who agreed with at least one of the three stigmatizing statements such as “Most people living with HIV do not care if they infect other people”, “People living with HIV should feel ashamed of themselves”, “People get infected with HIV because they engage in irresponsible behaviours” and disagree with the fourth statement “Women living with HIV should be allowed to have babies if they wish” were considered to have some level of stigmatizing opinion about PLHIV [22]. These responses served as a numerator for the stigmatizing opinion calculation while the denominator was based on the number of healthcare staff who answered at least one of these statements [22].”

6. To what extent are the HIV stigma measurements consistent with other measures of HIV stigma in other settings/countries?

In terms of the measurement tool:

As mentioned above, we used an HIV-related stigma scale that had been used on a multitude of other countries. We also validated the tool for use in Kazakhstan before we used it and have referenced an earlier publication that describes this process. Our findings are consistent with research in other settings that have looked at HIV-related stigma in the healthcare setting. We have added the following text to address this comment:

Lines 163-168. “The original standardized assessment tool, validated in six countries (China, Dominica, Egypt, Kenya, Puerto Rico, and St. Christopher & Nevis), was employed to measure stigma through respondents’ opinions about PLHIV. The tool demonstrated strong reliability in the original validation, with a Cronbach's alpha of 0.78 [21]. For the current study, we revalidated the tool in Kazakh and Russian, achieving excellent internal consistency, with a Cronbach’s alpha of 0.86, indicating its high reliability for the study context.”

In terms of the findings, we have also added the following:

Lines 402-410. “Numerous studies have explored HIV-related stigma in healthcare, primarily focusing on the experiences of PLHIV [6, 15-18]. However, fewer studies have systematically investigated this issue among healthcare workers, including specific clinic staff such as physicians, nurses, and office managers. Despite this, existing literature consistently indicates moderate to high levels of stigma towards PLHIV within healthcare settings [13, 17, 18]. For instance, a study conducted among healthcare providers in Alabama and Mississippi of the United States (US) found comparable percentages of stigma, with 89% of respondents at urban healthcare centers and 91% at rural clinics demonstrating at least one stigmatizing attitude [25]. Similar findings have been observed in other countries such as Nigeria, China, Poland, and Iran [26-31].”

Thank you again for all the valuable comments and questions!

Comments to the Author

Reviewer #1

Comment 1:

(Line 394-395/lines 406-409) “For instance, a study conducted among healthcare providers in the Deep South US found comparable percentages of stigma, with 89% of respondents at urban healthcare centers and 91% at rural clinics demonstrating at least one stigmatizing attitude [25].”

Specify the context in which the study was conducted. Change “Deep South US” to “among healthcare staff in Alabama and Mississippi of the United States (US)”.

Thank you for your suggestion. We have made the requested change to specify the context of the study. The revised text now reads as follows. Lines 406-409:

"For instance, a study conducted among healthcare providers in Alabama and Mississippi of the United States (US) found comparable percentages of stigma, with 89% of respondents at urban healthcare centers and 91% at rural clinics demonstrating at least one stigmatizing attitude [24]."

Comment 2:

(Line 387) Rephrase “stigmatizing opinions about PLHIV” to Stigmatizing attitudes towards PLHIV.

Thank you. We have rephrased it now on lines 387-389:

“Those who reported experience working with PLHIV within the last 12 months and those with longer years of work in healthcare in general were less likely to hold stigmatizing attitudes towards PLHIV”

Comment 3:

Throughout the discussion section, it would be clearer and more reader-friendly if the either authors first state the key results or findings of the current study before delving into a comparison with previous studies. Presenting the results upfront ensures the readers understand the ma

---

## [Decision Letter · Decision Letter 2]

4 Jun 2025

PONE-D-24-31521R2"I think they are infected because of their ignorance and lack of responsibility": a mixed-methods study on HIV-related stigma in the healthcare system in Kazakhstan.PLOS ONE

Dear Dr. Iskakova,

Thank you for submitting your manuscript to PLOS ONE. After careful consideration, we feel that it has merit but does not fully meet PLOS ONE’s publication criteria as it currently stands. Therefore, we invite you to submit a revised version of the manuscript that addresses the points raised during the review process. Please submit your revised manuscript by Jul 19 2025 11:59PM. If you will need more time than this to complete your revisions, please reply to this message or contact the journal office at plosone@plos.org. Please include the following items when submitting your revised manuscript:

We look forward to receiving your revised manuscript.

Kind regards,

Joseph KB Matovu, Ph.D.

Academic Editor

PLOS ONE

**Journal Requirements:**

**Additional Editor Comments:**

1. The explanation about collecting data from more respondents than the estimated sample size needs to be further tightened. Yes, the researchers' argument that a big sample size improves the accuracy of the estimates and the study power. However, this does not justify interviewing more people than the estimated sample size. A note should be included in the "Discussion" section to state the implications of shooting over and above the estimated sample size and how this might be avoided in future studies.

Reviewers' comments:

Reviewer's Responses to Questions

**Comments to the Author**

1. If the authors have adequately addressed your comments raised in a previous round of review and you feel that this manuscript is now acceptable for publication, you may indicate that here to bypass the “Comments to the Author” section, enter your conflict of interest statement in the “Confidential to Editor” section, and submit your "Accept" recommendation.

Reviewer #1: All comments have been addressed

Reviewer #2: (No Response)

2. Is the manuscript technically sound, and do the data support the conclusions?

Reviewer #1: Yes

Reviewer #2: Yes

3. Has the statistical analysis been performed appropriately and rigorously? 

Reviewer #1: Yes

Reviewer #2: Yes

4. Have the authors made all data underlying the findings in their manuscript fully available?

Reviewer #1: Yes

Reviewer #2: No

5. Is the manuscript presented in an intelligible fashion and written in standard English?

Reviewer #1: Yes

Reviewer #2: Yes

6. Review Comments to the Author

**Reviewer #1: **Line 44: Rephrase “HIV-positive” to patients living with HIV, to avoid using stigmatizing statements that refer to PLHIV.

Line 130: “Mixed methods” is an approach, then 134 “sequential explanatory” is a research design. This should be clearly articulated.

Line 329 -330: Italicize the verbatim account of a Midwife for consistency.

A sample of 10 participants who participated in the qualitative phase were selected purposefully from those who had already completed a quantitative survey? Or they were not at all part of the first phase. If they were not part of the first phase a rational is needed to elucidate why a different sample was invited. However, if they were part, authors must explain in detail how they were recruited and selected for the second phase.

I have never seen the section that integrate quantitative results with qualitative findings, provided that this study was designed sequentially whereby qualitative findings had to explain quantitative results. Without such integration, this defeats the purpose of sequential explanatory design, as the essence of mixed methods is about combining insights. Therefore, some form of integration is essential.

**Reviewer #2:** I have a few edits:

For the qualitative phase, 10 participants were selected from the survey sample for the in-depth interviews. [delete the - For the qualitative phase, 10 participants were selected from the survey sample for in-depth interviews.]

• The paper-based questionnaires were self-administered to provide more privacy to the respondents. [Paper-based questionnaires were self-administered to provide more privacy to the respondents.]

• and disagree with the fourth statement [disagreed]

• then depending on new finding [findings]

• demonstrated protective effect [demonstrated a protective effect]

• while those have not seen [while those had…]

• The qualitative interview participants frequently expressed concerns about the transmission of HIV when they think of …[thought of]

• be careful in choosing sexual partners with females [be careful in choosing female sexual partners]

• They reported feeling of “discomfort” and “disgust” [They reported feelings…]

7. PLOS authors have the option to publish the peer review history of their article (what does this mean?). If published, this will include your full peer review and any attached files.

Reviewer #1: No

Reviewer #2: No

---

## [Author Response · Author response to Decision Letter 3]

3 Jul 2025

Manuscript PONE-D-24-31521

Response to Reviewers

Dear Reviewers and the Editorial Team,

We would like to thank you for your time and effort in reviewing our manuscript titled "I think they are infected because of their ignorance and lack of responsibility": A Mixed-Methods Study on HIV-Related Stigma in the Healthcare System in Kazakhstan.

In response to your suggestions and comments, we have made the necessary revisions and ensured that all edits are marked for your convenience. Specifically, we have submitted two versions of the revised manuscript: one with tracked changes and one clean version without them. In the current document, our responses are provided in blue font along with updated line numbers for ease of reference.

Once again, thank you for your time, dedication, and expertise.

Sincerely,

Corresponding author,

Balnur Iskakova, PhD

School of Public Health

Asfendiyarov Kazakh National Medical University

Additional Editor Comments:

The explanation about collecting data from more respondents than the estimated sample size needs to be further tightened. Yes, the researchers' argument that a big sample size improves the accuracy of the estimates and the study power. However, this does not justify interviewing more people than the estimated sample size. A note should be included in the "Discussion" section to state the implications of shooting over and above the estimated sample size and how this might be avoided in future studies.

Thank you for pointing this out. Following your suggestion, we have added this issue on lines 484-489:

It is also important to mention that although the estimated sample size for this study was 380, 448 eligible respondents were ultimately included. This was due to a high response rate within the data collection period. All eligible respondents were included to improve statistical power and representativeness. However, we acknowledge this as a deviation from the original protocol and recommend that future studies define stopping criteria in advance to manage such scenarios.

Review Comments to the Author

Reviewer #1:

Line 44: Rephrase “HIV-positive” to patients living with HIV, to avoid using stigmatizing statements that refer to PLHIV.

We agree with this point since terminology is important and evolving rapidly. Therefore, we have replaced all instances of “HIV-positive patients” with “Patients living with HIV” throughout the manuscript to use person-first, non-stigmatizing language.

Line 130: “Mixed methods” is an approach, then 134 “sequential explanatory” is a research design. This should be clearly articulated.

Thank you for this comment. We have now clarified the distinction between the general approach and the specific design, and revised the text accordingly as follows on lines 131-137:

“A mixed-methods approach was used in this study to construct a better understanding of HIV-related stigma in primary healthcare. Specifically, we employed a sequential explanatory design, in which quantitative survey data were collected and analyzed first as the primary data source. Subsequently, qualitative in-depth interviews were conducted to complement and explain the survey findings. Lastly, we combined the findings from both phases in order to draw conclusions, focusing on how the qualiative data could better explain our quantative results. This design is illustrated in Fig 1.”

Line 329 -330/lines 340-341: Italicize the verbatim account of a Midwife for consistency.

Thank you for your observation. We have now italicized the verbatim quote for consistency. The revised text appears as follows:

“They are the victims of those who sell drugs for money. It is a profit to someone else. I feel bad for them.” Midwife, 60 years old

A sample of 10 participants who participated in the qualitative phase were selected purposefully from those who had already completed a quantitative survey? Or they were not at all part of the first phase. If they were not part of the first phase a rational is needed to elucidate why a different sample was invited. However, if they were part, authors must explain in detail how they were recruited and selected for the second phase.

Thank you for the question. We understand the need for clarity regarding recruiting participants for the qualitative phase. As previously mentioned in the manuscript:

“For the qualitative phase, 10 participants were selected from the survey sample for the in-depth interviews. Using a purposive sampling strategy, we selected a mix of clinical and non-clinical personnel, including nurses, doctors, psychologists, and social workers.”

To further clarify, we have now expanded the description of the recruitment process in the Methods section of the manuscript. The following text has been added on lines 151-157:

"During the quantitative survey, participants were asked if they were willing to be contacted for a potential follow-up in-depth interview. Respondents who agreed to this provided a phone number for further contact. All identifying information such as full names or addresses was not collected to ensure confidentiality. From this pool of respondents who had consented to be re-contacted, 10 participants representing diverse clinical and non-clinical roles were purposively selected and invited to participate in one-on-one interviews at a time convenient for them."

I have never seen the section that integrate quantitative results with qualitative findings, provided that this study was designed sequentially whereby qualitative findings had to explain quantitative results. Without such integration, this defeats the purpose of sequential explanatory design, as the essence of mixed methods is about combining insights. Therefore, some form of integration is essential.

Thank you for this comment. We agree that the integration of quantitative and qualitative findings is essential in a sequential explanatory design. To address this, we have added a dedicated paragraph in the Discussion section to explicitly integrate the findings. The revised section reads as follows on lines 401-415:

“These findings were further explored in the qualitative component of our study, which confirmed and contextualized these patterns. While the survey revealed widespread endorsement of negative beliefs toward PLHIV and key affected populations, the interviews clarified the moral and cultural judgments underlying these views. Although many participants expressed empathy for the psychosocial challenges faced by PLHIV, this empathy was often conditional and did not extend to those perceived as leading “immoral” lifestyles. Sex workers and MSM were frequently described in explicitly stigmatizing terms, suggesting that stigma is driven not only by fear of infection but also by entrenched moral biases. Furthermore, fear of occupational exposure, reported by the majority of survey participants, was a recurring theme in interviews. Despite acknowledging the availability of protective measures, many participants described this fear as “natural” and emphasized the need to take extra precautions when treating PLHIV. This reflects an underlying anxiety that may contribute to differential or discriminatory clinical practices. The interviews also revealed limited direct contact with PLHIV, and that participants expressed a willingness to engage in more interactive and experience-based training to address HIV-related stigma and discrimination.”

Reviewer #2: I have a few edits:

For the qualitative phase, 10 participants were selected from the survey sample for the in-depth interviews. [delete the - For the qualitative phase, 10 participants were selected from the survey sample for in-depth interviews.]

Thank you. We corrected it now as follows:

“For the qualitative phase, 10 participants were selected from the survey sample for in-depth interviews.”

The paper-based questionnaires were self-administered to provide more privacy to the respondents. [Paper-based questionnaires were self-administered to provide more privacy to the respondents.]

Thank you. We corrected it now as follows:

“Paper-based questionnaires were self-administered to provide more privacy to the respondents.”

and disagree with the fourth statement [disagreed]

Thank you. We corrected it to the past tense now:

“The number of people who agreed with at least one of the three stigmatizing statements such as “Most people living with HIV do not care if they infect other people”, “People living with HIV should feel ashamed of themselves”, “People get infected with HIV because they engage in irresponsible behaviors” and disagreed with the fourth statement “Women living with HIV should be allowed to have babies if they wish” were considered to have some level of stigmatizing opinion about PLHIV [22].”

• then depending on new finding [findings]

Thank you. We have corrected it now to a plural version of the word “finding” on linew 234-236

“The interviews were coded following the section “Opinions about PLHIV” first deductively then depending on new findings, inductive codes were added.”

• demonstrated protective effect [demonstrated a protective effect]

Thank you. We corrected it now as follows on lines 286-287:

“Multivariable logistic regression models which adjusted for the covariates given in Table 2 demonstrated a protective effect…”

• while those have not seen [while those had…]

Thank you, this part is corrected now on line 288

“while those who had not seen a patient with HIV within the last 12 months had higher odds of holding stigmatizing opinions about PLHIV (AOR=3.31; 95% CI =1.73, 6.35; p=0.001)”

• The qualitative interview participants frequently expressed concerns about the transmission of HIV when they think of …[thought of]

We corrected this now on lines 317-318

“The qualitative interview participants frequently expressed concerns about the transmission of HIV when they thought of PLHIV, emphasizing the importance of preventive measures.”

• be careful in choosing sexual partners with females [be careful in choosing female sexual partners]

Thank you, this part is corrected now on lines 354-355

“According to the data, participants had more negative attitudes towards females, saying that men should be careful in choosing female sexual partners.”

• They reported feeling of “discomfort” and “disgust” [They reported feelings…]

Thank you, this part is corrected now on line 367

“They reported feelings of “discomfort” and “disgust” leading to unwillingness to be in contact with such individuals.”

---

## [Editor Report · Decision Letter 3]

13 Aug 2025

"I think they are infected because of their ignorance and lack of responsibility": a mixed-methods study on HIV-related stigma in the healthcare system in Kazakhstan.

PONE-D-24-31521R3

Dear Dr. Iskakova,

We’re pleased to inform you that your manuscript has been judged scientifically suitable for publication and will be formally accepted for publication once it meets all outstanding technical requirements.

Kind regards,

Joseph KB Matovu, Ph.D.

Academic Editor

PLOS ONE
---

## [Editor Report · Acceptance letter]

PONE-D-24-31521R3

PLOS ONE

Dear Dr. Iskakova,

I'm pleased to inform you that your manuscript has been deemed suitable for publication in PLOS ONE. Congratulations! Your manuscript is now being handed over to our production team.

Kind regards,

on behalf of

Dr. Joseph KB Matovu

Academic Editor

PLOS ONE